# On the role of VP3-PI3P interaction in birnavirus endosomal membrane targeting

**Flavia A Zanetti**[1†], **Ignacio Fernandez**[2†], **Eduard Baquero**[2], **Pablo Guardado-Calvo**[2], **Andres Ferrino-Iriarte**[2], **Sarah Dubois**[3], **Etienne Morel**[3], **Victoria Alfonso**[4], **Milton Osmar Aguilera**[5], **María E Celayes**[5], **Luis Mariano Polo**[5], **Laila Suhaiman**[6], **Vanesa V Galassi**[6,7], **Maria V Chiarpotti**[6,7], **Carolina Allende-Ballestero**[8], **Javier M Rodriguez**[8], **Jose R Castón**[8], **Diego Lijavetzky**[9], **Oscar Taboga**[4], **María I Colombo**[5], **Mario Del Pópolo**[6,7], **Félix A Rey**[2], **Laura Ruth Delgui**[5,7]*

[1]Instituto de Ciencia y Tecnología "Dr. Cesar Milstein", Consejo Nacional de Investigaciones Científicas y Técnicas (CONICET), Buenos Aires, Argentina; [2]Institut Pasteur, Université Paris Cité, Structural Virology Unit, Paris, France; [3]Université Paris Cité, INSERM UMR-S1151, CNRS UMR-S8253, Institut Necker Enfants Malades, Paris, France; [4]Instituto de Agrobiotecnología y Biología Molecular (IABIMO), Instituto Nacional de Tecnología Agropecuaria (INTA), Consejo Nacional de Investigaciones Científicas y Técnicas (CONICET), Buenos Aires, Argentina; [5]Instituto de Histología y Embriología de Mendoza, Universidad Nacional de Cuyo (UNCuyo), Consejo Nacional de Investigaciones Científicas y Técnicas (CONICET), Centro Universitario, Mendoza, Argentina; [6]Instituto Interdisciplinario de Ciencias Básicas (ICB), Consejo Nacional de Investigaciones Científicas y Técnicas (CONICET), Mendoza, Argentina; [7]Facultad de Ciencias Exactas y Naturales, Universidad Nacional de Cuyo (UNCuyo), Mendoza, Argentina; [8]Department of Structure of Macromolecules, Centro Nacional de Biotecnología (CNB-CSIC), Madrid, Spain; [9]Instituto de Biología Agrícola de Mendoza, Universidad Nacional de Cuyo (UNCuyo), Consejo Nacional de Investigaciones Científicas y Técnicas (CONICET), Mendoza, Argentina

**\*For correspondence:**
ldelgui@mendoza-conicet.gob.ar

[†]These authors contributed equally to this work

## eLife Assessment

Zanetti et al use **convincing** biophysical and cellular assays to investigate the interaction of the birnavirus VP3 protein with the early endosome lipid PI3P. The study provides **valuable** insights and will be of interest to virologists. In future studies, it would be interesting to demonstrate that VP3-PIP3P is a specific interaction and not a general interaction with other PIPs.

**Abstract** Birnaviruses are a group of double-stranded RNA (dsRNA) viruses infecting birds, fish, and insects. Early endosomes (EE) constitute the platform for viral replication. Here, we study the mechanism of birnaviral targeting of EE membranes. Using the Infectious Bursal Disease Virus (IBDV) as a model, we validate that the viral protein 3 (VP3) binds to phosphatidylinositol-3-phosphate (PI3P) present in EE membranes. We identify the domain of VP3 involved in PI3P-binding, named P2 and localized in the core of VP3, and establish the critical role of the arginine at position 200 ($R_{200}$), conserved among all known birnaviruses. Mutating $R_{200}$ abolishes viral replication. Moreover, we propose a two-stage modular mechanism for VP3 association with EE. Firstly, the carboxy-terminal region of VP3 adsorbs on the membrane, and then the VP3 core reinforces the membrane

engagement by specifically binding PI3P through its P2 domain, additionally promoting PI3P accumulation.

## Introduction

Birnaviruses belong to the *Birnaviridae* family composed of nonenveloped icosahedral dsRNA viruses that infect a wide range of vertebrate and invertebrate hosts. The prototypical and best-characterized family member, the IBDV, is the causative agent of Gumboro disease, a highly contagious immunosuppressive disease that affects young chickens (*Gallus gallus*) causing significant damage to the poultry industry (*Delmas et al., 2019*).

Birnaviruses are unconventional dsRNA viruses, as the virions lack a '*core*,' i.e., an inner protein layer containing the genome, which instead forms a filamentous ribonucleoprotein complex (RNP) with the multifunctional protein VP3, the RdRp VP1, and the dsRNA genome segments (*Luque et al., 2009b*). IBDV contains a single ~70 nm diameter T=13 l icosahedral capsid surrounding a polyploid bipartite dsRNA genome (segments A and B, 3.2, and 2.8 kbp, respectively) (*Luque et al., 2009a*). The X-ray crystal structure of the capsid revealed that VP2 is the only component. VP2 has two main domains, a 'shell' domain homologous to the coat proteins of +sRNA such as the T=3 nodaviruses, and a 'tower' domain, like those present in the T=13 middle-layer protein of dsRNA viruses of the order *Reovirales* (*Coulibaly et al., 2005*). VP3 is a 257-residues long polypeptide that plays multiple roles during the viral life cycle. In the mature virion, VP3 is a component of the RNPs, where it is associated with the dsRNA (*Luque et al., 2009b*). During viral assembly, VP3 acts as a scaffold protein via electrostatic interactions between basic residues of pVP2 (the precursor form of VP2), and acidic residues at the VP3 carboxy-terminal (Ct) region (*Saugar et al., 2010*). Additionally, VP3 interacts with VP1 (*Lombardo et al., 1999*). VP3's multifunctional properties might be mediated by its capacity to form oligomers (*Casañas et al., 2008*), or by its intrinsically disordered regions. The X-ray crystal structure of the central region of the IBDV VP3 (residues 82–222) showed that it consists entirely of α-helices connected by loops which can be divided into two structural domains linked by a flexible hinge. The 36 Ct residues of the protein, a highly hydrophilic region rich in charged amino acids and proline residues, were predicted to be disordered, so it was excluded from the crystallized construct. The 81 amino-terminal (Nt) region of VP3 was cleaved during the crystallization process and was absent in the crystal (*Casañas et al., 2008*). The X-ray crystal structure indicated the presence of two positively charged domains, termed Patch 1 (P1) and Patch 2 (P2) (*Valli et al., 2012*). Both patches are composed of four discontinuous, positively charged residues forming two exposed patches on the surface of the protein. P1 was demonstrated to be involved in the dsRNA binding-activity of VP3 (*Valli et al., 2012*).

Previous studies on the subcellular localization of the IBDV replication complex (RC) have shown that IBDV replication components are localized to EE (*Delgui et al., 2013*). Further analysis revealed that VP3 associates with the cytosolic leaflet of EE membranes via P2, with a critical functional role in the virus life cycle (*Gimenez et al., 2018*). Additionally, the presence of PI3P in EE membranes was demonstrated to be a key host factor for VP3 association and the consequent establishment of IBDV RCs on EE membranes (*Gimenez et al., 2021*).

Here, we study the mechanism mediating the birnaviral targeting of EE membranes, focusing on the VP3-PI3P interaction. Using biophysical approaches in cell-free in vitro systems, we demonstrate that PI3P constitutes the cellular partner for VP3 association to EE membranes. By introducing charge-reversal point mutations on VP3 P2, we establish the contribution of each of the four positively charged residues composing P2, with a key role of the arginine at position 200 ($R_{200}$), a residue conserved across all known birnaviruses. Using a reverse genetic system for IBDV, we show that mutating $R_{200}$ abolishes viral replication. Finally, by using molecular simulations, we discuss a potential two-stage modular mechanism for VP3 association with EE. We show that, in the first step, the Ct flexible tail of VP3 adsorbs to the membrane by electrostatic interactions due to its positive charge. Then, in the second stage, the VP3 core reinforces the membrane engagement by recruiting PI3P molecules scattered in the EE membrane to the P2 region, promoting a stable VP3-PI3P interaction.

## Results

### Biophysical characterization of VP3 binding to PI3P

We set up a liposome co-flotation assay to assess the binding of VP3 to PI3P in membranes. We expressed and purified His-tagged VP3 full-length (His-VP3 FL) and confirmed its identity by Western blot using specific anti-VP3 and anti-His antibodies. Since we observed four different bands for the purified protein, we confirmed their VP3-identity by mass spectrometry (*Figure 1—figure supplement 1*). We prepared two populations of liposomes: 'liposomes PI3P(-)' and 'liposomes PI3P(+)'. We confirmed that these preparations were suitable for evaluating the interaction with PI3P by monitoring co-flotation of a recombinant His-2xFYVE domain (<u>F</u>ab1p, <u>Y</u>OTB, <u>V</u>ac1p, and <u>E</u>EA1), followed by immunoblot detection of the His-tag. FYVE domains are highly homologous cysteine-rich domains of 70–80 amino acids present in around 30 human proteins that are known to participate in vesicular sorting or endocytosis through direct binding to PI3P from EE (*Burd and Emr, 1998*; *Gaullier et al., 1998*; *Simonsen et al., 1998*; *Stenmark et al., 1996*; *Figure 1A*). We observed specific binding of His-VP3 FL to liposomes PI3P(+), evidenced by a significant difference in the ability to bind to both liposome populations (*Figure 1B*). To test His-VP3 FL PI3P-binding specificity, we performed a co-flotation assay using liposomes where PI3P was replaced by 1,2-dioleoyl-sn-glycero-3-phosphate ('liposomes PA') or [1,2-dioleoyl-sn-glycero-3-phospho-(1'-myo-inositol)] ('liposomes PI') at the same molar ratio. PA contains a $POH_4^-$ group also present in the polar head of PI3P but lacks the inositol ring, while PI has the inositol ring but is not phosphorylated. We observed that His-VP3 FL bound to liposomes PI3P(+), but not to liposomes PA or PI, reinforcing the notion that a phosphoinositide is required since neither a single negative charge nor an inositol ring are sufficient to promote VP3 binding to liposomes (*Figure 1—figure supplement 2*).

Two additional biophysical approaches were implemented. First, we prepared liposomes PI3P(-) and PI3P(+), and incubated them with His-2xFYVE or His-VP3 FL that had been pre-bound to Ni-NTA [nickel (II) nitrilotriacetic acid] gold particles. Subsequently, we inspected the preparations by cryo-electron microscopy (cryo-EM). Gold particles were not observed on liposomes PI3P(-) incubated with either protein (*Figure 1C*, left panel). We observed gold particles decorating liposomes PI3P(+) in the presence of both, His-2xFYVE and His-VP3 FL (*Figure 1C*, right panel). Second, we performed bio-layer interferometry (BLI) experiments using purified His-VP3 FL, and liposomes PI3P(-) and PI3P(+) (*Shah and Duncan, 2014*). In the BLI experiments, the protein is immobilized on the tip of Ni-NTA sensors while the liposomes remain in solution, allowing to monitor the molecular interactions in real-time (*Kairys et al., 2019*). Purified His-Streptavidin and His-2xFYVE were used as negative and positive controls, respectively. A specific liposome PI3P(+) dose-dependent binding of His-2xFYVE and His-VP3 FL was observed (*Figure 1D*).

Previous studies indicated that the P2 is responsible for PI3P binding on EE membranes (*Gimenez et al., 2021*; *Gimenez et al., 2018*). Yet the Ct region of VP3 is also highly enriched in positively charged residues. We generated a computational model of FL VP3 using AlphaFold2 (*Jumper et al., 2021*), which predicted a disordered Ct region (pLDTT of 42.6) (*Figure 1E*). Thus, we hypothesized that both conditions, the positive electrostatic surface charge and the disordered character of the VP3 Ct, might be contributing to the binding to the surface of EE. To test this hypothesis, we made a VP3 construct bearing a Ct truncation (His-VP3 ΔCt) and used it in co-flotation assays and BLI experiments. We did not detect an interaction of this mutant with PI3P(-) nor PI3P(+) liposomes (*Figure 1F and G*). Taken together, our results validate the role of PI3P in the interaction with VP3 and underscore the significance of the Ct domain in mediating this interaction.

### Role of VP3 P2 in the association of VP3 with the EE membrane

Inspection of the VP3 crystal structure showed that P2 residues ($K_{157}$, $R_{159}$, $H_{198}$, and $R_{200}$) form a positive-charged groove on the protein surface (*Figure 2A and B*). We prepared VP3 point mutants with reversed charge in a pcDNA vector backbone, giving rise to pcDNA VP3 $K_{157}D$, $R_{159}D$, $H_{198}D$, and $R_{200}D$. The calculated electrostatic potential of P2 for each of these mutants is shown in *Figure 2C*. Firstly, we used QM7 cells transiently overexpressing VP3 P2 wild-type (WT), P2 (all reversed), or the four single mutation of P2 to quantitatively assess their cellular distribution. We observed a punctuated distribution of VP3 P2 WT, $K_{157}D$, $R_{159}D$, and $H_{198}D$. In contrast, the specific indirect immunofluorescence (IIF) signal of VP3 P2 all reversed and $R_{200}D$ showed a diffuse cytosolic distribution (*Figure 2D*, panels framed in red). Second, we assessed the distribution of the VP3 constructs in QM7

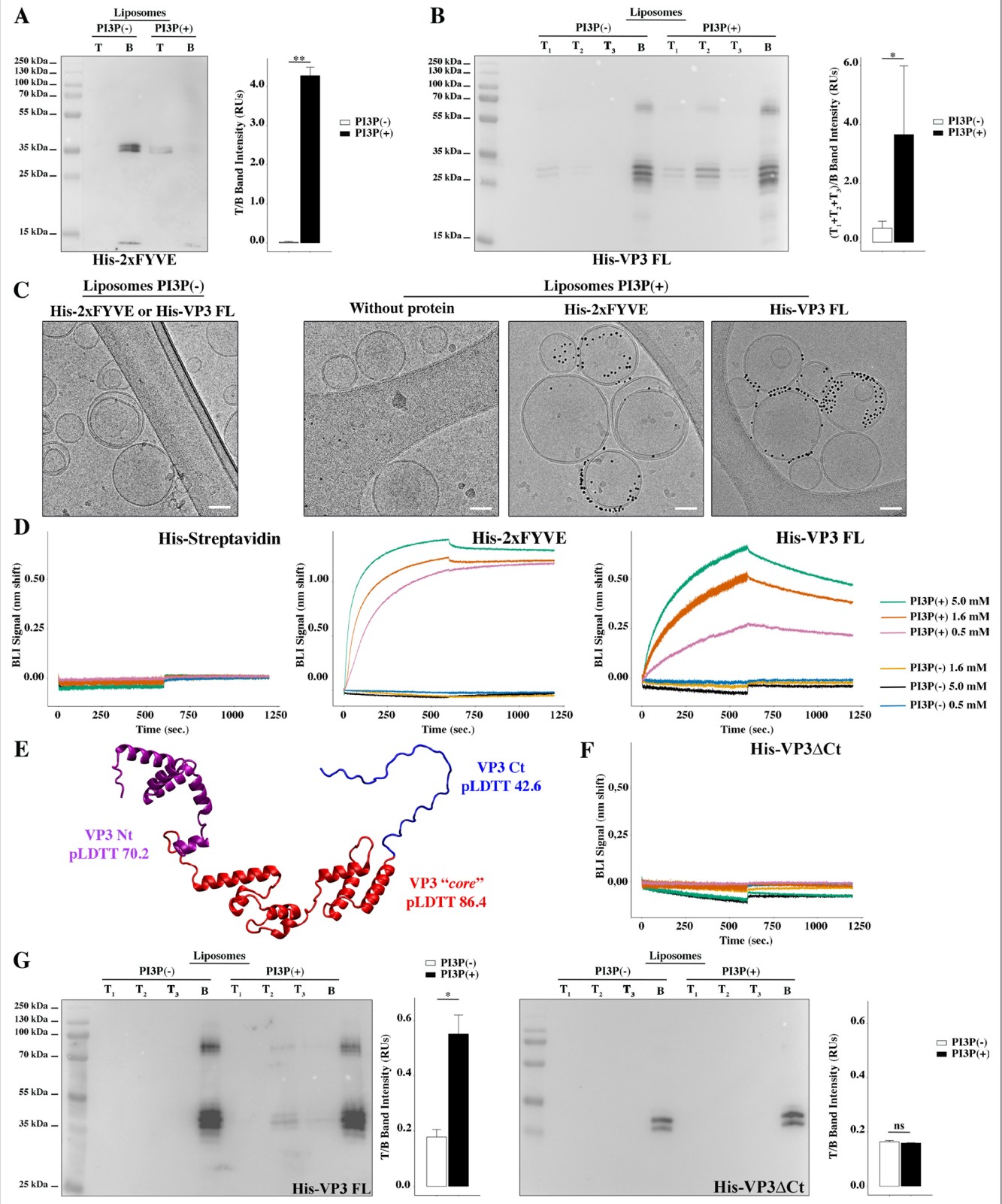

**Figure 1.** Biophysical characterization of viral protein 3 (VP3) binding to phosphatidylinositol-3-phosphate (PI3P). (**A**) Immunoblots of the top (**T**) and bottom (**B**) fractions from a liposome PI3P(-) or PI3P(+) OptiprepTM co-floatation assay indicating that His-2xFYVE protein (~35 kDa) specifically binds to liposome PI3P(+). Results are representative of three independent experiments. The bar plot represents the intensity of T/B bands for each liposome preparation. Significant differences (**p<0.01) as determined by one-way ANOVA with Tukey's HSD test.(**B**) Immunoblots of the three top (T₁, T₂, and

*Figure 1 continued on next page*

*Figure 1 continued*

$T_3$) and bottom (**B**) fractions from a liposome PI3P(-) or PI3P(+) OptiprepTM co-floatation assay indicating that His-VP3 FL protein (~32 kDa) specifically binds to liposome PI3P(+). Results are representative of three independent experiments. The bar plot represents the intensity of ($T_1+T_2+T_3$)/B bands for each liposome preparation. Significant differences (*$p<0.05$) as determined by one-way ANOVA with Tukey's HSD test. (**C**) Liposomes were vitrified and only representative cryo-electron microscopy images are shown. Left panel: liposomes PI3P(-) incubated even with the Ni-NTA gold particles reagent alone or pre-incubated with His-2xFYVE of His-VP3 FL. Right panel: liposomes PI3P(+) control (without protein), or incubated with His-2xFYVE- or His-VP3 FL-Ni-NTA gold particles showing gold particles decorating the membrane of the liposomes when His-2xFYVE or His-VP3 FL were present. The bar represents 50 nm. (**D**) Binding of His-Streptavidin (negative control), His-2xFYVE (positive control), or His-VP3 FL to three different concentrations of liposomes PI3P(-) or PI3P(+). Association and dissociation sensorgrams measured by bio-layer interferometry (BLI), showing the specific interaction of His-2xFYVE and His-VP3 FL with liposomes PI3P(+) in a dose-dependent manner, as indicated. (**E**) Cartoon representation of AlphaFold2 prediction of VP3 FL. Red region corresponds to the '*core*' region of the protein present in the experimental X-ray crystallographic model obtained by Casañas and coworkers PDB: 2R18 (*Casañas et al., 2008*). Regions not present within the PDB are colored in violet and blue representing the Nt and the Ct of VP3, respectively. pLDTT values lower than 50 are a strong predictor of disorder. (**F**) Binding of His-VP3 ΔCt to three different concentrations of liposomes PI3P(-) or PI3P(+). Sensorgrams measured by BLI, showing the absence of binding to either liposomes when the VP3 lacks the Ct region blue in (**E**). (**G**) Immunoblots of the three top ($T_1$, $T_2$, and $T_3$) and bottom (**B**) fractions from a liposome PI3P(-) or PI3P(+) OptiprepTM co-floatation assay of His-VP3 FL protein (~32 kDa) (positive control, left panel) or His-VP3 ΔCt protein (~28 kDa) (right panel) showing the lack of VP3 ΔCt binding to both liposomes. Results are representative of three independent experiments. The bar plot represents the intensity of ($T_1+T_2+T_3$)/B bands for each liposome preparation. Significant differences (*$p<0.05$; ns $p>0.05$) as determined by one-way ANOVA with Tukey's HSD test.

The online version of this article includes the following source data and figure supplement(s) for figure 1:

**Source data 1.** Original membranes corresponding to *Figure 1A, B and G*.

**Source data 2.** Individual files corresponding to the original membranes from *Figure 1A, B and G*.

**Figure supplement 1.** His-viral protein 3 (VP3) Full length (FL) purification and characterization.

**Figure supplement 1—source data 1.** Original Coomasie R. Blue-stained polyacrylamide gel and western blot membranes corresponding to *Figure 1—figure supplement 1B and C*.

**Figure supplement 1—source data 2.** Individual files corresponding to the original Coomasie R. Blue-stained polyacrylamide gel and western blot membranes from *Figure 1—figure supplement 1B and C*.

**Figure supplement 2.** His-viral protein 3 (VP3) full length (FL) phosphatidylinositol-3-phosphate (PI3P)-binding specificity.

**Figure supplement 2—source data 1.** Original western blot membranes corresponding to *Figure 1—figure supplement 2*.

**Figure supplement 2—source data 2.** Individual files corresponding to the original western blot membranes from *Figure 1—figure supplement 2*.

cells transiently overexpressing EGFP-Rab5. Rab5 regulates the homotypic tethering and fusion of EEs, and it is considered a genuine marker of EEs (*Gorvel et al., 1991*). As observed in *Figure 2E*, and the quantitative colocalization analysis (*Figure 2—figure supplement 1*), VP3 P2 WT, $K_{157}D$, $R_{159}D$, and $H_{198}D$ showed a marked colocalization with EGFP-Rab5. In contrast, the VP3 P2 all reversed and $R_{200}D$-derived IIF signal indicated a cytosolic distribution (*Figure 2E*, panels framed in red). It is important to mention that the differences in co-localization shown in *Figure 2—figure supplement 1* are small and not significant, but still show a trend that correlates with observations depicted in *Figure 2D*. Taken together, these results suggest that $R_{200}$ in the P2 patch bears a critical role in the association of VP3 with EE membranes.

## VP3 P2 mediates VP3-PI3P association to EE membranes

To further assess the contribution of P2 residues to PI3P binding, and to strengthen the observations made on cells transiently overexpressing EGFP-Rab5, we set up two additional cell biology assays. One consists of co-transfecting cells with EGFP-2xFYVE and VP3 constructs, and detecting them by confocal laser scanning microscopy (CLSM). The second one consists of assessing the ability of purified GST-2xFYVE, which recognizes endogenous PI3P, to co-localize with VP3. We employed QM7 cells transiently co-overexpressing EGFP-2xFYVE and VP3 P2 WT, P2 all reversed, or the four-point mutants to evaluate their colocalization. As shown in *Figure 3A*, we observed a punctuated distribution with co-localization with EGFP-2xFYVE for VP3 P2 WT while for VP3 P2 all reversed the IIF signal indicated a cytosolic distribution of the protein with the loss of colocalization with EGFP-2xFYVE (*Figure 3A*, red framed panels on the upper two rows). Regarding the point mutants, we observed a loss of punctuated distribution and EGFP-2xFYVE co-localization for VP3 FL $R_{200}D$ (*Figure 3A*, red framed panels on the bottom row). Then, we used the GST-2xFYVE purified domain to recognize endogenous PI3P in fixed cells (*Nascimbeni et al., 2017*). GST-2xFYVE was used as the 'primary antibody,' which was labeled with a fluorescent anti-GST antibody. As an additional control in these

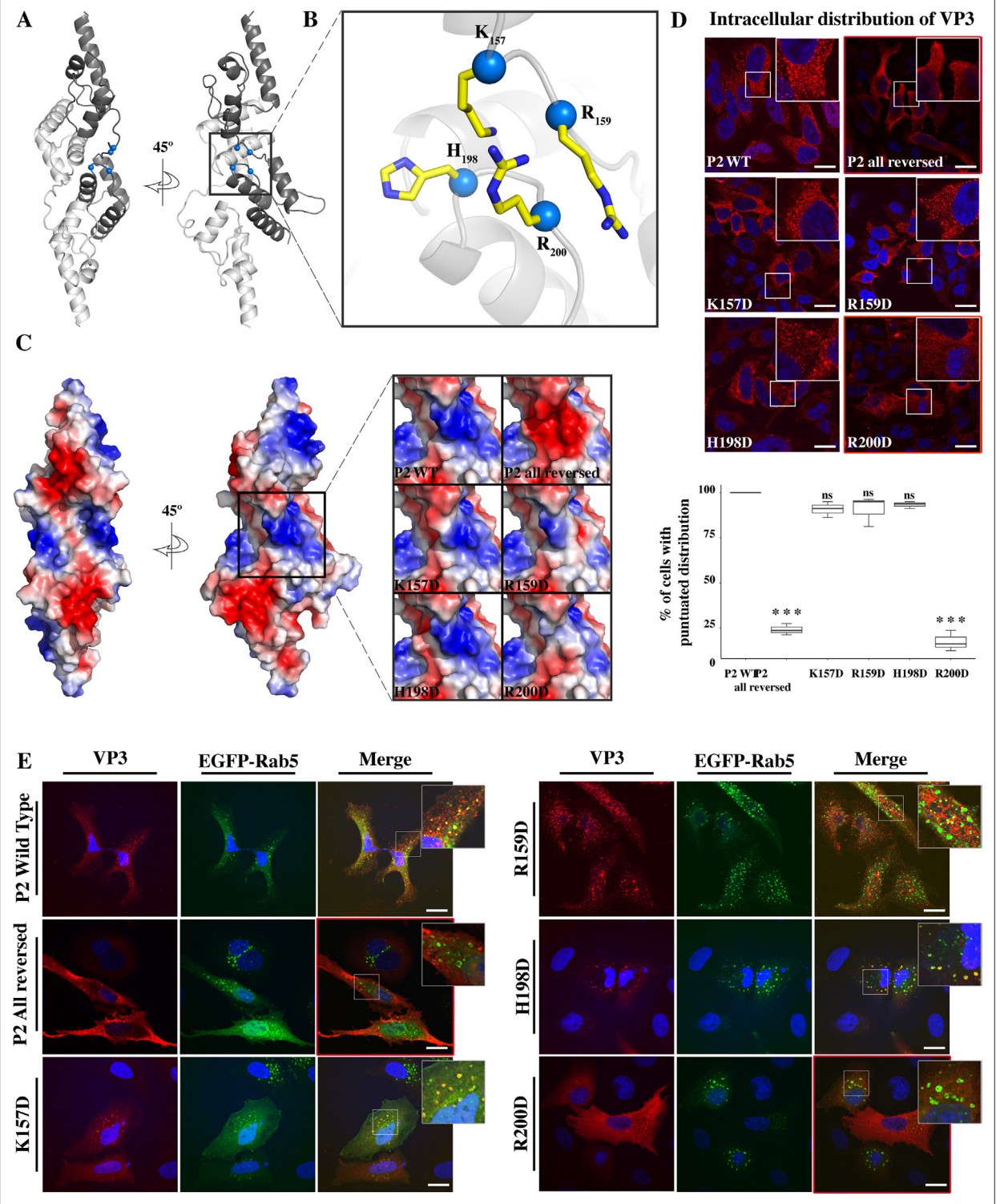

**Figure 2.** Viral protein 3 (VP3) P2 involvement in the association of VP3 with the early endosomes (EE) membrane. (**A**) The cartoon representation of the VP3 dimer PDB 2R18 (*Casañas et al., 2008*) shows each protomer in different shades of gray. The blue balls depict the residues defining the P2 region. (**B**) The close-up of P2 region showing residues $K_{157}$, $R_{159}$, $H_{198}$, and $R_{200}$. (**C**) Electrostatic potential mapped on the surface of the structure of a VP3 dimer in the same orientations as in (**A**) structures. The close-up shows the impact of P2 residue mutations on the electrostatic potential of the binding site. P2 wild-type (WT) corresponds to the 'UniProt code'. The color-coded electrostatic surface potential of VP3 was drawn using PyMol (blue positive, red negative). (**D**) QM7 cells transfected with pcDNA VP3 FL (P2 WT), P2 (all reversed), or the four-point mutants ($K_{157}D$, $R_{159}D$, $H_{198}D$, and $R_{200}D$) and immunostained with anti-VP3 showing the distribution of each protein (upper panel). Images were captured using a Confocal Laser Scanning

*Figure 2 continued on next page*

*Figure 2 continued*

Microscopy and then the percentage of cells with punctated fluorescent signal were determined for each protein (lower panel). The red signal shows the VP3 distribution and the blue one shows the nuclei, which were Hoestch-stained. The data were normalized to the P2 WT protein. The box plot represents the percentage of cells with punctuated distribution of VP3. Significant differences (***p<0.001; ns p>0.05.) as determined by one-way ANOVA with Tukey's HSD test. (**E**) QM7 cells co-transfected with pEGFP-Rab5 and pcDNA VP3 FL (P2 WT), P2 (all reversed), or the four-point mutants ($K_{157}D$, $R_{159}D$, $H_{198}D$, and $R_{200}D$) and immunostained with anti-VP3 showing the distribution of each protein. Representative images, captured using a Confocal Laser Scanning Microscopy are shown where green signal represents Rab5 distribution and the red signal that of VP3. Nuclei were Hoestch-stained and are blue. White bar-scales represent 20 mm. VP3 P2 (all reversed) and $R_{200}D$ depict a cytosolic distribution of the proteins. Quantification of the co-localization of the different VP3 proteins and EGFP-Rab5 is shown in *Figure 2—figure supplement 1*.

The online version of this article includes the following figure supplement(s) for figure 2:

**Figure supplement 1.** VP3-EGFP-Rab5 co-localization quantification.

experiments, we immunostained the EE with antibodies anti-EEA1. As shown in *Figure 3B*, VP3 P2 WT demonstrated a conspicuous PI3P-bearing EEA1 positive endosomes location. In addition, we observed a similar distribution for VP3 $K_{157D}$, $R_{159}D$, and $H_{198}D$ and a drastic change in VP3 FL $R_{200}D$ mutant, which appeared utterly cytosolic, losing its capacity to bind to PI3P-bearing EE membranes (*Figure 3B*, red framed panel on the lowest row). These results further support the role of $R_{200}$ in this interaction.

## VP3 $R_{200}$ is crucial for binding to PI3P

To validate this, we devised a liposome co-flotation assay using a recombinant His-VP3 FL $R_{200}D$ mutant. When assayed in parallel with His-2xFYVE, we observed that His-VP3 FL $R_{200}D$ lost binding to PI3P(+) liposomes (*Figure 4A*). To confirm that the lack of binding was not due to misfolding of the mutant, we compared the circular dichroism spectra of His-VP3 FL and His-VP3 FL $R_{200}D$ proteins, without detecting significant differences (*Figure 4B*). To assess its relevance in IBDV infectivity, we resorted to an IBDV reverse genetics system, based on a modification of the system described by Qi and coworkers (*Qi et al., 2007*). The full-length sequences of the IBDV RNA segments A and B, flanked by a hammerhead ribozyme at the 5'-end and the hepatitis delta ribozyme at the 3'-end, were expressed under the control of an RNA polymerase II promoter within the plasmids pCAGEN.Hmz. SegA.Hdz (SegA) and pCAGEN.Hmz.SegB.Hdz (SegB). For this specific experiment, we generated a third plasmid, pCAGEN.Hmz.SegA.$R_{200}D$.Hdz (SegA.$R_{200}D$), harboring a mutant version of segment A cDNA containing the $R_{200}D$ substitution. Then, QM7 cells were transfected with the plasmids SegA, SegB, or Seg.$R_{200}D$ alone (as controls) or with a mixture of plasmids SegA +SegB (wild-type situation) or SegA.$R_{200}D$+SegB (mutant situation). At 8 hr post-transfection (p.t.), when the new viruses have been able to assemble out of the two segments of RNA, the cells were recovered and re-plated onto fresh non-transfected cells to reveal the presence of infective viruses. At 72 hr post-plating, the generation of foci forming units (FFUs) was revealed by Coomassie staining. As expected, single-transfections of SegA, SegB, or Seg.$R_{200}D$ did not produce FFUs. As shown in *Figure 4C*, the transfection of SegA +SegB produced detectable FFUs (the three circles in the upper panel) while no FFUs (the three circles in the lower panel) were detected after the transfection of SegA.$R_{200}D$+SegB (*Figure 4C*). Given the relevance of VP3 $R_{200}$, we assessed its conservation among members of the *Birnaviridae* family. For this, a multiple sequence alignment of VP3 reference sequences was performed (*Figure 4—figure supplement 1A*). As shown, IBDV VP3 has 32–39% identity to the other selected homologues (*Figure 4—figure supplement 1B*). A closer look at the VP3 P2 revealed the complete conservation of $R_{200}$ (*Figure 4D*). Taken together, these findings show that VP3 $R_{200}$ plays an important role in mediating the VP3-PI3P interaction, highlighting the active participation of EE membranes during the infectious cycle of birnaviruses.

## Electrostatic forces mediate VP3 membrane binding

Computer simulations based on coarse-grain models were performed to assess the binding free-energy and adsorption mechanism of three protein variants: VP3 Ct, VP3 ΔNt, and VP3 FL. The VP3 Ct region, with a positive electrical charge of +5, was modeled as an intrinsically disordered region according to the AlphaFold2 prediction (*Figure 1E*). VP3 ΔNt encompasses residues 82–257 and is electroneutral. This variant was chosen primarily because it retains the structural data from the crystallized protein. Similarly to His-VP3 FL, His-VP3 ΔNt retains the ability to bind to liposomes PI3P(+)

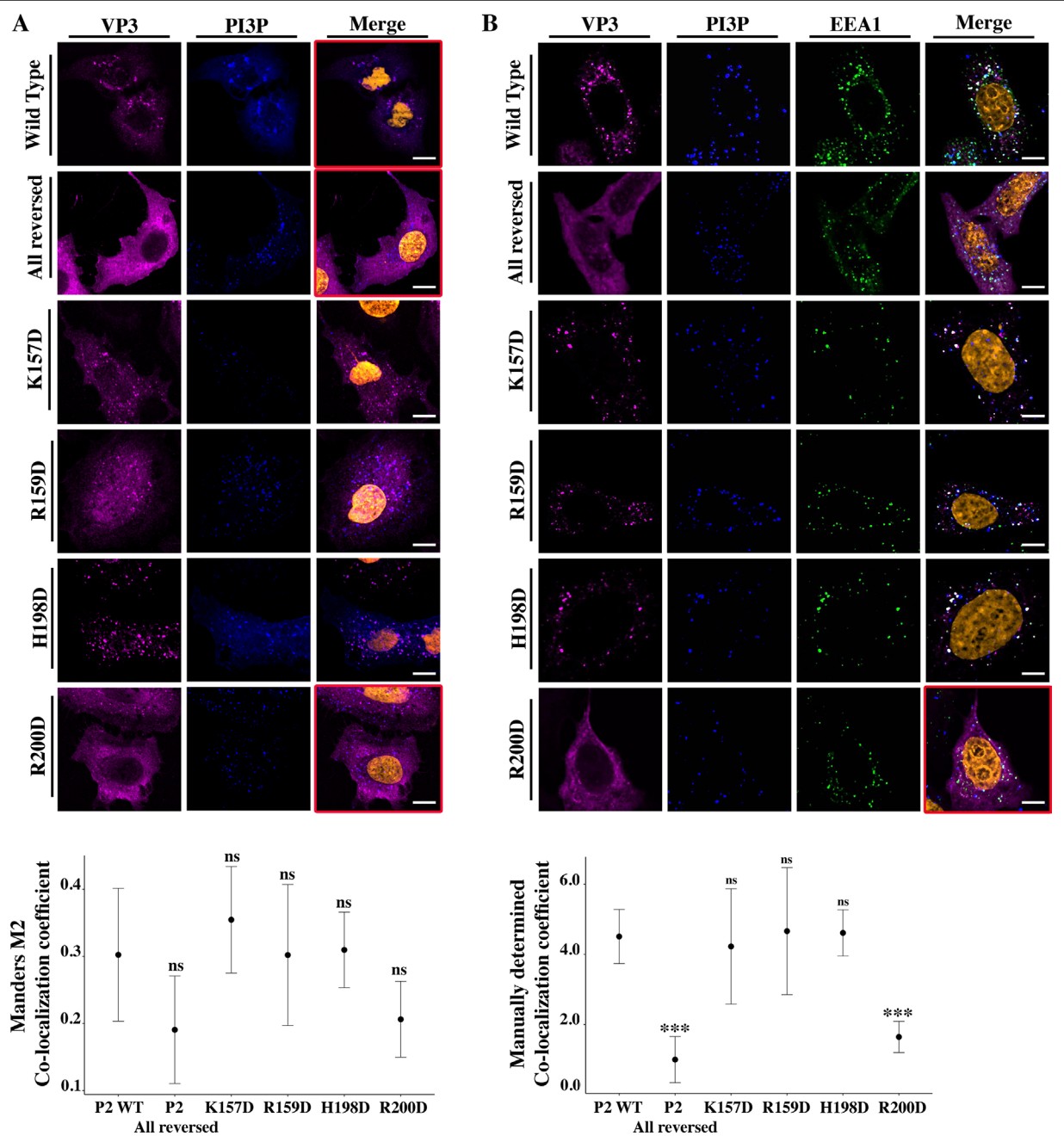

**Figure 3.** Viral protein 3 (VP3) P2 involvement in the association between VP3 and early endosomes (EE) phosphatidylinositol-3-phosphate (PI3P). (**A**) QM7 cells co-transfected with the PI3P biosensor pEGFP-2xFYVE and pcDNA VP3 FL (P2 WT), P2 (all reversed), or the four-point mutants ($K_{157}D$, $R_{159}D$, $H_{198}D$, and $R_{200}D$) and immunostained with anti-VP3 showing the distribution of each protein. Representative images captured using a Confocal Laser Scanning Microscopy are shown where blue signal represents FYVE distribution and the magenta signal that of VP3. Nuclei were Hoestch-stained and are orange (upper panels). White bar-scales represent 20 mm. VP3 P2 (all reversed) and $R_{200}D$ depict a cytosolic distribution of the proteins with a significant lower co-localization coefficient. The dot plot in the lower panel depicts the co-localization coefficient for each protein determined as explained in the Materials and methods section. Significant differences (ns $p>0.05$) as determined by one-way ANOVA with Tukey's HSD test. (**B**) QM7 cells were transfected with the pcDNA VP3 FL (P2 WT), P2 (all reversed), or the four-point mutants ($K_{157}D$, $R_{159}D$, $H_{198}D$, and $R_{200}D$). The cells were fixed and then the GST-2xFYVE purified peptide and anti-VP3 antibodies were used to recognize endogenous PI3P and VP3, respectively. Additionally, anti-EEA1 antibodies were used to stain the endosomes. GST-2xFYVE was labelled with a fluorescent anti-GST antibody (blue signal), anti-VP3, and anti-EEA1 antibodies with fluorescent secondary antibodies (magenta and green signals, respectively), and Hoestch-stained nuclei in orange. White bar-scales represent 10 mm. VP3 P2 (all reversed) and $R_{200}D$ depict a cytosolic distribution of the proteins with a significant lower co-localization coefficient. The dot plot in the lower panel depicts the co-localization coefficient for each protein determined as explained in the Materials and methods section. Significant differences (***$p<0.001$; ns $p>0.05$) as determined by one-way ANOVA with Tukey's HSD test.

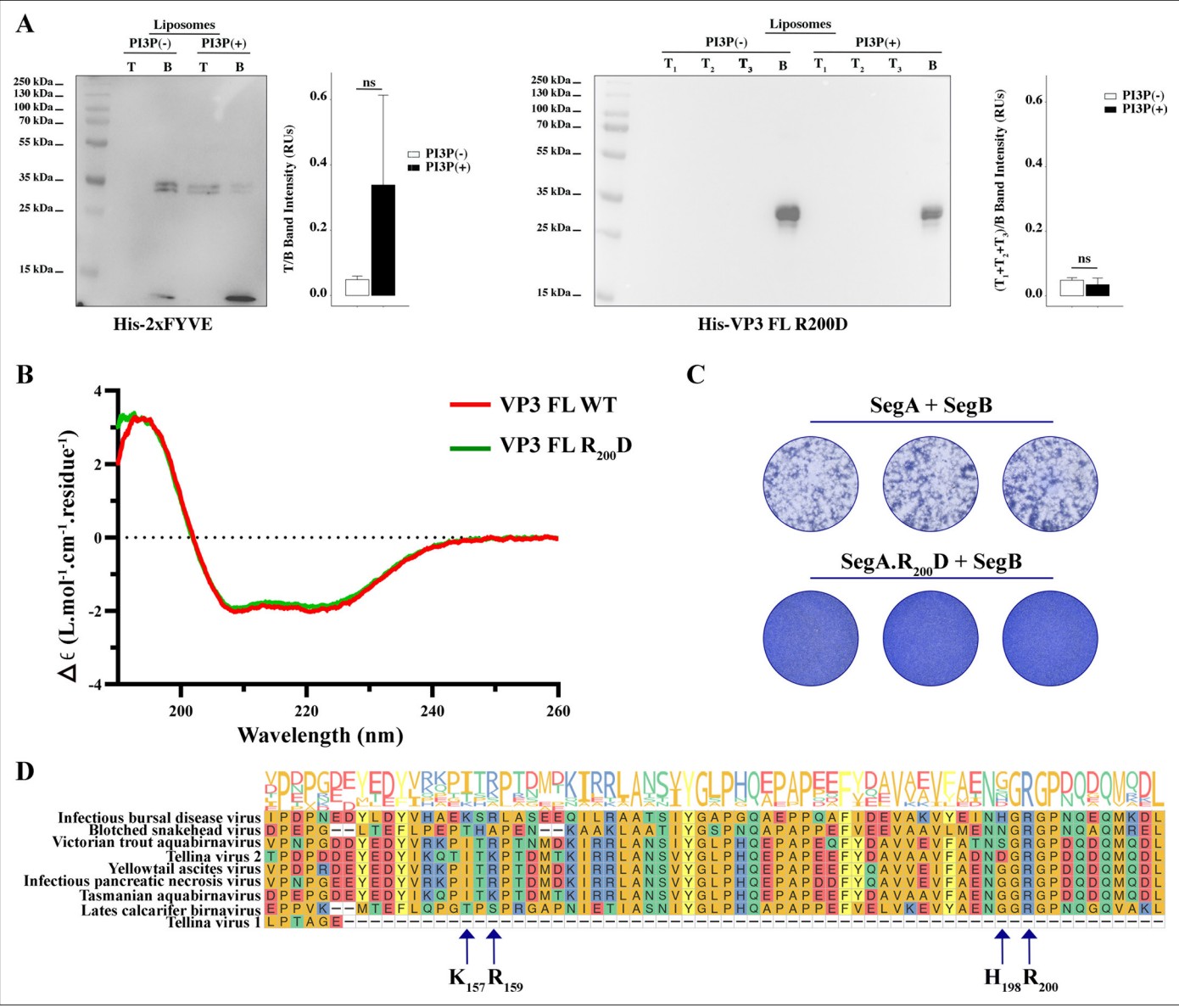

**Figure 4.** Biophysical characterization of viral protein 3 (VP3) full length (FL) $R_{200}D$ binding to phosphatidylinositol-3-phosphate (PI3P). (**A**, left panel). Immunoblots of the top (T) and bottom (**B**) fractions from a liposome PI3P(-) or PI3P(+) OptiprepTM co-floatation assay indicating that His-2xFYVE protein (~35 kDa) specifically binds to liposome PI3P(+). The bar plot represents the intensity of T/B bands for each liposome preparation. Significant differences (ns p>0.05) as determined by one-way ANOVA with Tukey's HSD test. (**A**, right panel). Immunoblots of the three top ($T_1$, $T_2$, and $T_3$) and bottom (**B**) fractions from a liposome PI3P(-) or PI3P(+) OptiprepTM co-floatation assay indicating that His-VP3 FL $R_{200}D$ protein (~35 kDa) does not bind to liposome PI3P(-) nor PI3P(+). The bar plot represents the intensity of ($T_1+T_2+T_3$)/B bands for each liposome preparation. Significant differences (ns p>0.05) as determined by one-way ANOVA with Tukey's HSD test. (**B**) Far-UV CD spectra of His-VP3 FL (red line) or His-VP3 FL $R_{200}D$ (green line). Spectral acquisitions at 50 nm/min with 0.1 nm steps at 1 s integration time, with a bandwidth of 1 nm were performed four times for the samples as well as for the buffer. The measurements were carried out with constant nitrogen gas flux of 10 ml/min. Acquisitions were averaged and buffer baseline was subtracted with Spectra Manager (JASCO). No smoothing was applied. CDtoolX was used to zero between 255–260 nm and to calibrate the signal amplitude from the fresh CSA signal (***Miles and Wallace, 2018***). Data are presented as delta epsilon (Δε) per residue (L.mol[-1].cm-[1].residue[1]) calculated using the molar concentration of protein and number of residues. (**C**) QM7 cells were grown in M24 multi-well plate for 12 hr to approximately 90–95% confluency and then 800 ng of plasmids were transfected [(SegA +SegB) or (SegA.R$_{200}$D+SegB)] in triplicate. At 8 hr post-transfection (p.t.) the supernatants were discarded, and the monolayers were recovered for further plating on M6 multi-well plates containing non transfected QM7 cells. Avicel RC-591 (FMC Biopolymer) was added to the M6 multi-well plates. 72 hr p.i., the monolayers were fixated and stained with Coomassie R250 for revealing the foci forming units. (**D**) Partial view of amino acid alignment of the VP3 protein for nine reference members of *Birnaviridae* family. Multiple sequence alignment was performed with Clustal OMEGA (v1.2.4) (***Sievers et al., 2011***) implemented at EMBL's European Bioinformatics Institute (***Thakur et al., 2024***) (The complete alignment is shown in ***Figure 4—figure supplement 1A***). Alignment visualization was done with the R *ggmsa*

*Figure 4 continued on next page*

*Figure 4 continued*

package (*Zhou et al., 2022*) in with assistance from the RStudio software (*RStudio Team, 2020*). Amino acids are colored according to their side-chain chemistry. Protein sequence logos annotation is displayed on top of the amino acid alignment. For facilitating the view IBDV VP3 142–210 portion is shown. The black arrow indicates the $K_{157}$, $R_{159}$, $H_{198}$, and $R_{200}$.

The online version of this article includes the following source data and figure supplement(s) for figure 4:

**Source data 1.** Original membranes corresponding to *Figure 4A*.

**Source data 2.** Individual files corresponding to the original membranes from *Figure 4A*.

**Figure supplement 1.** Bioinformatic analysis.

---

(*Figure 5—figure supplement 1*). Finally, VP3 FL is the full-length monomeric VP3, using the Alpha-Fold2 predicted structure with an overall electrostatic charge of –3. We employed the Molecular Theory (MT) approach to obtain adsorption free-energy profiles, PMF(z), at different salt concentrations (*Chiarpotti et al., 2021*; *Ramírez et al., 2019*). The membrane was modeled as a rigid dielectric slab with 5% of its surface occupied by ternary ionizable groups. *Figure 5* (panels A-C) shows the PMF(z), where z represents the distance between the center of mass of the protein and the center of the membrane, at 50 and 150 mM NaCl. As observed in *Figure 5A*, the positively charged VP3 Ct shows the highest affinity for the membrane due to the deprotonation of the acidic lipids at pH 8, resulting in a net negative charge on the membrane surface (*Figure 5—figure supplement 2*). However, the electroneutral construct VP3 ΔNt has a lower binding energy than VP3 Ct, and also VP3 FL shows some degree of membrane binding, despite both the protein and the membrane being negatively charged, as evidenced by the shallow minimum in the PMF.

The binding of the three protein constructs was strongly affected by electrostatics, as shown in *Figure 5D*, where the adsorption energy is plotted as a function of the concentration of salt. The effect of salt concentration is most significant for VP3 Ct, but it is also noticeable for VP3 ΔNt and VP3 FL. Also, the interfacial concentration of each construct was higher at lower ionic strengths, as depicted in *Figure 5—figure supplement 3*. The weak binding energy of VP3 FL to the anionic membrane under physiological conditions (~150 mM NaCl, pH 7) was consistent with the experimental observation that VP3 FL did not bind to liposomes PA and PI, which are anionic at the pH of the experiments. However, those same experiments showed that VP3 FL did bind to PI3P(+) liposomes (*Figure 1B–C*), despite the MT calculations predicting weak binding. Taken together, our results suggest that specific VP3-PI3P interactions enhance the binding of VP3 to PI3P-containing EE membranes, while exhibiting weak and non-specific electrostatic attractions to an otherwise generic anionic membrane. *Figure 5—figure supplement 4* illustrates the distribution of cationic residues of VP3 near the membrane, indicating that the Ct fragment drives the non-specific binding. To identify the electrical charge of different protein regions, *Figure 5—figure supplement 5* shows a map of the electrostatic potential around VP3 FL.

Molecular Dynamics (MD) simulations based on the MARTINI model were employed to further investigate the mechanism by which VP3 Ct and VP3 ΔNt approach the membrane surface. We chose VP3 ΔNt over VP3 FL because VP3 ΔNt has a closer resemblance to the well-established experimental crystal structure. Additionally, the adsorption free energy profiles predicted by MT for both VP3 ΔNt and VP3 FL are notably similar (compare panels B and C of *Figure 5*). Unlike the MT model, MARTINI considers both electrostatic and van der Waals forces (*de Jong et al., 2013*). To mimic the liposome PI3P(+) composition used in the co-flotation assays, the lipid bilayer was modeled with a molar ratio of 64:31:5 for the lipids DOPE, DOPC, and PI3P, respectively. DOPE and DOPC are electroneutral, PI3P is anionic. The MD simulations revealed that both VP3 Ct and VP3 ΔNt settled on the membrane surface after 120 ns, as shown in *Figure 6—figure supplement 1*. *Figure 6A–D* shows a temporal sequence of configurations during the adsorption of VP3 ΔNt. The protein approached the membrane via the Ct fragment and remained adsorbed during the rest of the 500 ns trajectory. VP3 Ct exhibited the same behavior, as demonstrated in *Figure 6—figure supplements 1 and 2*. Notably, VP3 Ct and VP3 ΔNt did not detach from the surface during these MD simulations, suggesting that the adsorption energies calculated by MT were underestimated and represent a lower bound, as van der Waals interactions are absent in the MT model. Also, *Figure 6E* highlights the location of the four positive residues that make up the P2 region, showing their proximity to the membrane surface.

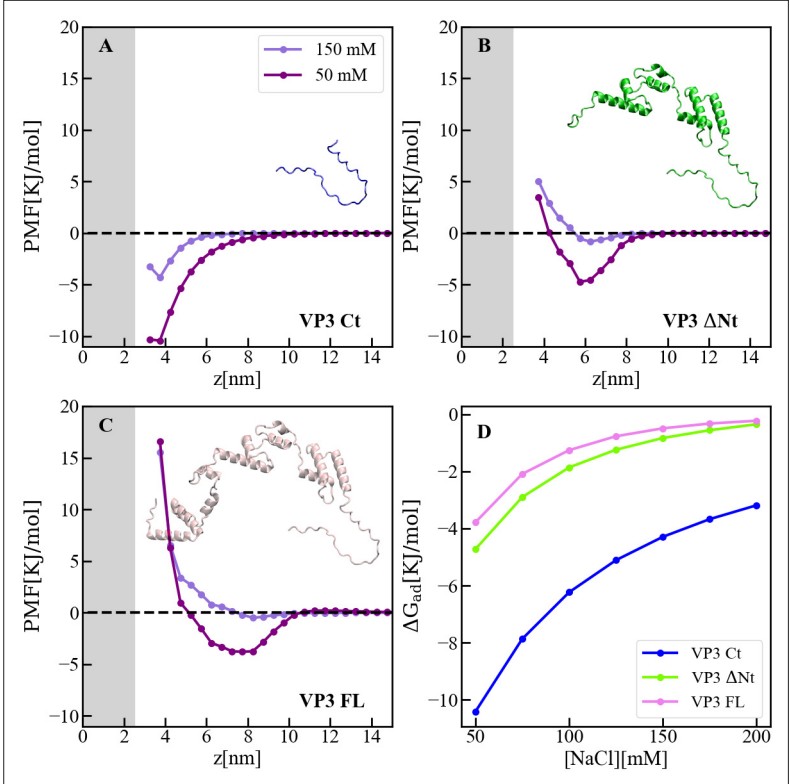

**Figure 5.** Adsorption of viral protein 3 (VP3) constructs to phosphatidylinositol-3-phosphate (PI3P)(+) model membranes. (**A–C**). Adsorption free-energy profiles, PMF(z),for VP3 Ct, VP3 ΔNt, and VP3 FL at 50 and 150 mM of NaCl. (**D**) Adsorption free energy (ΔG$_{ad}$), computed from the minimum of PMF(z), versus concentration of NaCl. In all cases, the solution pH is 8, and the concentration of protein is 1 μM. The gray areas represent the volume excluded by half the membrane (cis hemilayer, z>0). The membrane surface contains 5% titratable groups, representing PI3P. Each group has three acidic moieties, one with a pKa of 2.5 and the others with 6.5. At 150 mM NaCl and pH 8, more than 90% of the acidic groups are deprotonated (**Figure 5—figure supplement 2**).

The online version of this article includes the following source data and figure supplement(s) for figure 5:

**Figure supplement 1.** Biophysical characterization of His-viral protein 3 (VP3) DNt binding to phosphatidylinositol-3-phosphate (PI3P).

**Figure supplement 1—source data 1.** Original western blot membranes corresponding to *Figure 5—figure supplement 1A*.

**Figure supplement 1—source data 2.** Individual files corresponding to the original western blot membranes from *Figure 5—figure supplement 1A*.

**Figure supplement 2.** Titration curve of the membrane.

**Figure supplement 3.** Interfacial concentration of viral protein 3 (VP3) constructs.

**Figure supplement 4.** Distribution of charged and aromatic residues of viral protein 3 (VP3).

**Figure supplement 5.** Map of electrostatic potential around viral protein 3 (VP3) full length (FL).

## Recruitment of PI3P modulates the membrane binding of VP3

We investigated the impact of VP3 Ct and VP3 ΔNt on the composition of the membrane near the binding point, and on the local curvature of PI3P(+) liposomes. We resorted to our MD simulations and evaluated the in-plane radial distribution function, g(r), between the center of mass of the proteins and the center of mass of PI3P molecules in the (cis) hemilayer, which is in direct contact with the protein. *Figure 6F* shows that VP3 Ct recruits PI3P within a~2 nm radius, where its local concentration in the cis hemilayer is up to 6 times greater than the average PI3P concentration (*Figure 6—figure supplement 3*). Also, despite having no net electrical charge, VP3 ΔNt induces a similar concentration enhancement, with PI3P accumulation occurring around the Ct fragment, as evidenced in *Figure 6— figure supplement 4*. In both cases, the recruitment of PI3P in the cis hemilayer is accompanied by

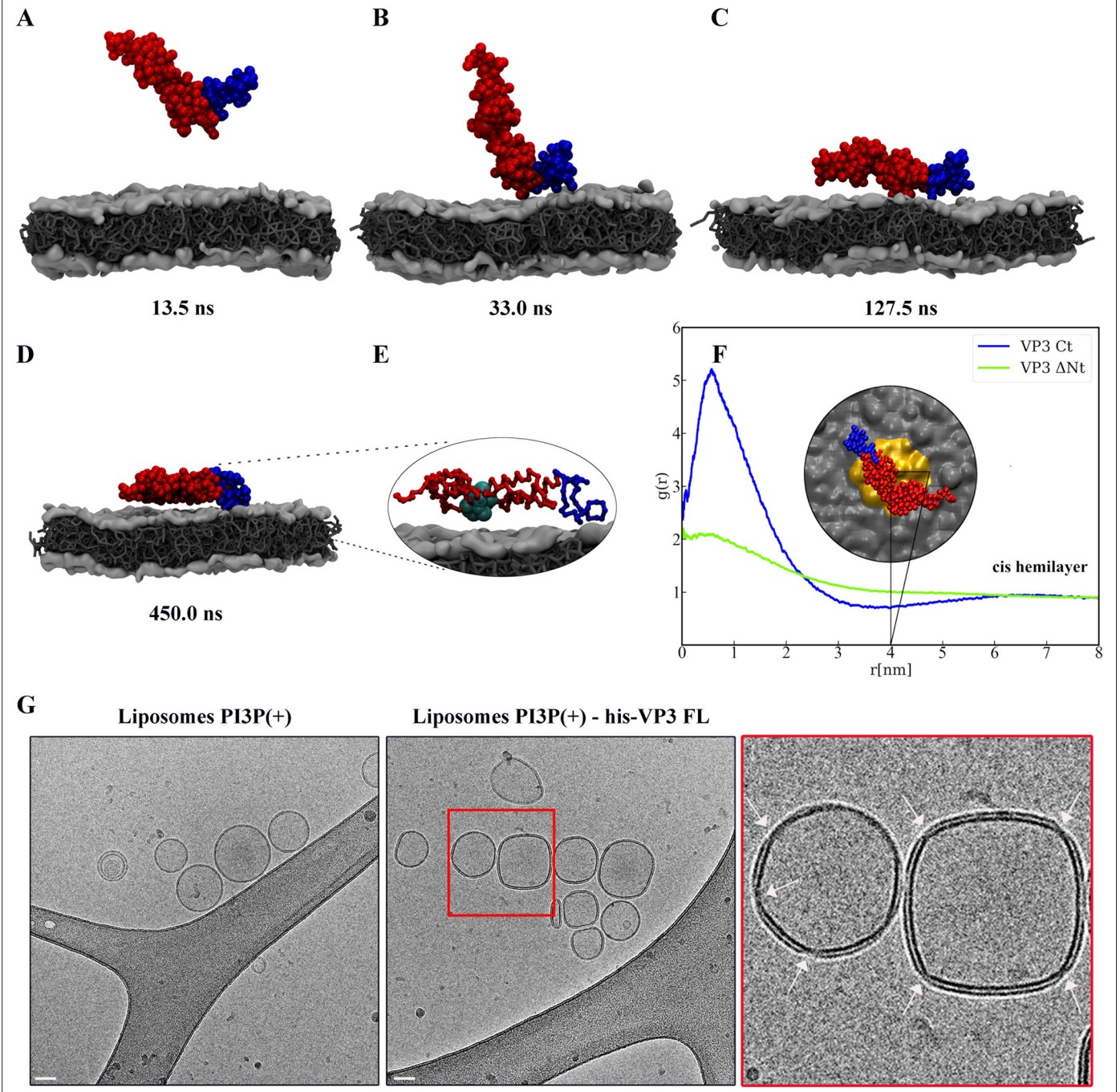

**Figure 6.** Viral protein 3 (VP) approaching a lipid bilayer, and distortion of the membrane. (**A–D**) Temporal sequence of configurations illustrating how VP3 ΔNt approaches the negatively charged membrane during a 500 ns molecular dynamics (MD) simulation. The membrane contains 1,2-dioleoyl-sn-glycero-3-phosphoethanolamine (DOPE),1,2-dioleoyl-sn-glycero-3-phosphocholine (DOPC), and phosphatidylinositol-3-phosphate (PI3P) in 64:31:5 molar ratio. The beads making the carboxy-terminal (Ct) fragment are colored blue. (**E**) Magnification of the protein configuration bound to the membrane, with P2 residues represented in cyan balls. (**F**) Radial distribution function, g(r), between the center of mass of VP3 ΔNt and the center of PI3P molecules in the cis hemilayer. g(r) greater than 1 implies local enhancement of PI3P concentration. The inset shows an upper view of VP3 ΔNt bound to the membrane, with the area within r=4 nm colored in orange. (**G**) Cryo-electron microscopy images of cryo-fixated liposomes PI3P(+) control (without protein, left panel), or incubated with His-VP3 FL showing the small pinches or localized thinnings in the bilayer of the liposomes when His-VP3 FL was present (middle panel). The bar represents 50 nm. The right panel represents an enlarged image of the red square on the middle panel. White arrows point to the small pinches or localized thinnings in the bilayer when His-VP3 FL was present.

The online version of this article includes the following figure supplement(s) for figure 6:

*Figure 6 continued on next page*

*Figure 6 continued*

**Figure supplement 1.** Distance between membrane and protein constructs computed by molecular dynamics (MD).

**Figure supplement 2.** Initial and final configurations of viral protein 3 (VP3) carboxy-terminal (Ct) on a lipid membrane.

**Figure supplement 3.** Lipids accumulation/depletion around viral protein 3 (VP3) carboxy-terminal (Ct).

**Figure supplement 4.** Lipids accumulation/depletion around the carboxy-terminal (Ct) fragment of viral protein 3 (VP3) DNt.

**Figure supplement 5.** Lipids accumulation/depletion around viral protein 3 (VP3) ΔNt and its mutants.

**Figure supplement 6.** Adsorption free energy of viral protein 3 (VP3) ΔNt and VP3 ΔNt mutants on PI3P(+) model membranes.

a local depletion of PI3P in the trans hemilayer, due to interhemilayer electrostatic repulsions. The effects on PI3P concentration, induced by VP3 ΔNt, result from a synergistic action between the positively charged Ct and the P2 region of the protein. Furthermore, when mutations are introduced to the P2 region - starting with $R_{200}D$ and eventually including the entire P2 domain - there is a gradual reduction in its contribution to the recruitment of PI3P (*Figure 6—figure supplement 5*). Molecular Theory calculations on these mutants also reveal that the binding free energy of both VP3 ΔNt P2 Mut and VP3 ΔNt $R_{200}D$ is consistently lower than that of VP3 ΔNt across all salt concentrations explored (*Figure 6—figure supplement 6*). This indicates a weakening of the interaction with the negatively charged membrane, concomitant with the loss of positively charged amino acids.

Finally, to examine the potential impact of VP3 on the lipid bilayer, we incubated liposomes PI3P(-) and PI3P(+) with purified His-VP3 FL as described previously. The samples were cryo-fixed and cryo-EM was used to observe the resulting structures. As shown in *Figure 6G*, His-VP3 FL distorts the membrane of liposomes containing PI3P, resulting in small pinches in the bilayer, detectable in 36% of the liposomes PI3P(+), while it is completely absent in PI3P(-) liposomes. Taken together, our findings suggest that the binding of VP3 to EE membranes has the potential to alter the local lipid composition by recruiting PI3P, which could then lead to the localized membrane pinches observed in *Figure 6G*.

## Discussion

Several viruses hijack the PI3P metabolism machinery to complete their infectious cycle. However, only a few cases of viral proteins that bind PI3P have been described. Equine infectious anemia virus (EIAV) is a member of the lentivirus subgroup of retroviruses and replicates in macrophages. Retroviral assembly and release are directed by the structural precursor polyprotein Gag (*Wills and Craven, 1991*). EIAV Gag localizes to both the cell interior and to the plasma membrane due to its high-affinity interaction with PI3P (*Fernandes et al., 2011*; *Puffer et al., 1998*; *Tanzi et al., 2003*). Moreover, mutation of $K_{49}$, a residue in the PIs-binding pocket of Gag inhibited the release of viral-like particles from the plasma membrane (*Fernandes et al., 2011*). Another example is the poxviruses, a family of cytoplasmic DNA viruses that also rely on intracellular membranes to develop their envelope, whose morphogenesis requires enzymes from the cellular phosphoinositide metabolic pathway (*McNulty et al., 2010*). For vaccinia virus (VACV), the prototypic poxvirus, Kolli and coworkers showed that the VACV H7 protein binds to PI3P and PI4P (*Kolli et al., 2015*). Similarly to our observations in IBDV VP3, the authors uncovered two regions of H7 essential for viral replication. One region is a positive surface patch centered on $K_{108}$, and the other region is the flexible Ct tail (*Kolli et al., 2015*).

Our biophysical and molecular simulation results suggest a specific but weak interaction of VP3 with PI3P-bearing membranes. In co-floatation assays, we observed that His-VP3 FL reaches the top of the gradient, where liposomes PI3P(+) are located, with a high proportion of protein still found at the bottom of the gradient. This phenomenon was not observed with the His-2xFYVE protein. Additionally, the slopes of the BLI curves show that His-VP3 FL dissociates faster than His-2xFYVE from liposomes PI3P(+). Computer simulations have also revealed a weak attractive interaction between the negatively charged VP3 and membranes that contain PI3P. VP3 bears multiple essential roles during the viral life cycle. In this context, our hypothesis is that an early, specific, but weak interaction with PI3P is required for the formation of RCs associated with EE and represents an advantage for the virus that counts with VP3 to go forward into the replication cycle.

We identified the domain of VP3 involved in PI3P-binding, and established the critical role played by the arginine residue at position 200 of VP3 ($R_{200}$). This arginine is conserved among all known birnaviruses. Mutating $R_{200}$ abolishes viral replication. As previously mentioned, VP3 is a multifunctional viral

protein. Therefore, it was initially possible that mutating $R_{200}$ might affect other functions beyond PI3P binding and viral replication. However, $R_{200}$ is not located in any of the protein regions associated with its other known functions. For instance, the VP3 dimerization domain, located in the second helical domain, involves 81 interprotomeric contacts across three helices, but $R_{200}$ does not participate in these contacts (*Casañas et al., 2008*). Also, VP3 has an oligomerization domain mapped within the 42 C-terminal residues of the polypeptide, i.e., the segment of the protein composed by the residues at positions 216–257 (*Maraver et al., 2003*); VP3's ability to bind RNA is facilitated by a region of positively-charged amino acids, identified as P1, which includes $K_{99}$, $R_{102}$, $K_{105}$, and $K_{106}$ (*Valli et al., 2012*). Furthermore, our findings indicate that the $R_{200}D$ mutant retains a folding pattern similar to the wild-type protein. All these lead us to conclude that the loss of replication capacity of $R_{200}D$ viruses results from impaired, or even loss of, VP3-PI3P interaction.

It has recently been reported that the RCs of IBDV exhibit liquid-liquid phase separation (LLPS) (*Reddy et al., 2022*). In LLPS, one or more biomolecules form a network of mild homo- and heterotypic interactions, resulting in the formation of a distinct liquid phase within a liquid medium (*Banani et al., 2017*). In light of our results, it is conceivable to hypothesize that viral interaction with the EE membranes acts as a first step in the RCs biogenesis, leading to the growth of biomolecular condensates with LLPS characteristics around EE membranes, as recently proposed (*Brodrick and Broadbent, 2023*). In support of this model, both the RNA-dependent RNA polymerase VP d the viral dsRNA associate to endosomes and to the LLPS condensates (*Delgui et al., 2013*; *Reddy et al., 2022*). Further experiments are needed in order to link the association of VP3 to EE membranes with biomolecular condensate formation.

## Evolutionary links between +sRNA and dsRNA viruses

Birnaviruses integrate the double-strand RNA Baltimore group (*Baltimore, 1971*), and the newly created *Riboviria* taxon (a realm), that includes all RNA viruses encoding RdRps (*Walker et al., 2021*). However, the organization of the birnaviral RdRp (*Gorbalenya et al., 2002*) and the structure of its capsid (*Coulibaly et al., 2005*) had already indicated strong links between birnaviruses and +sRNA viruses, placing them in a special branch of the dsRNA viruses that appears to have evolved independently of the others. The membrane association of their replication complexes is a further specificity of birnaviruses that links them more to the +sRNA viruses than to the other dsRNA viruses. Indeed, within the megataxonomy merging together Baltimores´ and Linnean classifications proposed by Koonin et al., the *Birnaviridae* family status is uncertain (*Koonin et al., 2020*).

## Materials and methods
### Cell lines

Sf9 cell line from *Spodoptera frugiperda* pupal ovarian tissue were purchased from Gibco, Thermo Fisher Scientific, and used for His-VP3 polypeptides production in the Bac-to-Bac Baculovirus Expression System. Quail muscle fibroblasts (QM7; ATCC CRL-1962) were purchased from the American Type Culture Collection, and used for cell biology assays. The absence of mycoplasma was periodically verified by using the MycoAlert Mycoplasma Detection Assays purchased from Lonza.

### Recombinant prokaryotic-expression plasmid

We used an *Escherichia coli*-based prokaryotic system to obtain the His-2xFYVE and GST-2xFYVE fusion proteins. For the first one, the plasmid encoding EGFP-2xFYVE, pEGFP-2xFYVE, generated by Gillooly and collaborators (*Gillooly et al., 2000*) was kindly provided by Harald A. Stenmark (Norwegian Radium Hospital, Oslo, Norway) to our group and used as a template. The 2xFYVE fingers consisted of duplicated domains of residues 147–223 of mouse hepatocyte growth factor-regulated tyrosine kinase substrate (Hrs) (*Komada and Kitamura, 1995*) separated by a QGQGS linker. The prokaryotic vector expressing 2xFYVE was obtained following standard PCR methods with the oligonucleotide pair 2xFYVE (*Supplementary file 1*). The PCR fragment was inserted into the pET32a(+), which incorporates the thioredoxin A (TrxA) gene, a 6xHis tag, and Thrombin site (Ts) within the amino-terminal domain of the 2xFYVE, and the correctness of the insertion was confirmed by sequencing. The complete fusion protein TrxA.His.Ts.2xFYVE (hereafter named His-2xFYVE) is 954 base pairs

(bp)-long and contains 317 residues (~35 kDa). The plasmid encoding GST-2xFYVE was kindly gifted by Harald A. Stenmark (Norwegian Radium Hospital, Oslo, Norway) to the group of Etienne Morel.

## Recombinant eukaryotic-expression plasmids

We used a *Baculovirus*-based eukaryotic system to obtain the His-VP3 FL, His-VP3 FL $R_{200}D$, and His-VP3 ΔCt fusion proteins. The construct of the recombinant baculovirus (rBV) His-VP3 FL was generated from IBDV VP3 FL protein strain Soroa, GenBank AF140705.1, residues 756–1012 (257 residues) from the polyprotein cloned in pcDNA vector. Thus, IBDV VP3 FL protein was subcloned from plasmid pcDNA VP3 FL into the pFast Bac HTb vector (Gibco BRL), which incorporates a 6xHis tag at the N-terminal of the recombinant protein. Briefly, VP3 FL coding sequence was amplified by PCR with the oligonucleotide pairs VP3.Fw and VP3.Rv incorporating *Bam*HI and *Eco*RI restriction sites, respectively (*Supplementary file 1*). The resulting DNA fragment containing the VP3 gene was digested with *Bam*HI and *Eco*RI and ligated to plasmid pFast Bac HTb restricted with the same enzymes, generating the plasmid pFast Bac-his-VP3 FL. This plasmid contained the VP3 gene fused to a histidine tag under the control of the polyhedrin promoter. The complete fusion protein His-VP3 FL is 861 bp-long and contains 286 residues (~32 kDa). The generation of rBV His-VP3 FL $R_{200}D$ was performed following the same procedure, but using the pcDNAVP3 FL $R_{200}D$ as a template for subcloning. The complete fusion protein His-VP3 FL $R_{200}D$ is 861 bp-long and contains 286 residues (~32 kDa). The generation of rBV His-VP3 Δ223–257 (His-VP3 ΔCt) was performed as follows. A 663 nucleotide DNA fragment containing a truncated version of the VP3 sequence encoding a polypeptide lacking the 36 carboxi-terminal residues was amplified by PCR from pcDNAVP3 FL with the oligonucleotide pairs VP3.Fw and VP3 Δ223–257.Rv incorporating *Bam*HI and *Eco*RI restriction sites, respectively (*Supplementary file 1*). The DNA fragment was digested with *Bam*HI and *Eco*RI and was ligated to plasmid pFast Bac HTb restricted with the same enzymes, generating the plasmid pFast Bac-his-VP3 Δ223–257, hereafter named His-VP3 ΔCt. All the resulting plasmids were subjected to nucleotide sequencing to assess the correctness of the inserted VP3 sequence, and they were then used to produce the corresponding rBVs by using the Bac-to-Bac system and by following the manufacturer's instructions (Invitrogen). Briefly, the recombinant plasmids were transformed into DH10Bac competent cells to obtain recombinant bacmids. The white colonies on culture plates were picked, confirmed by PCR, and used for Sf9 cell transfection by using Cellfectin II Reagent (Gibco). The transfected cells were cultured at 28 °C for 5 d to produce rBVs. Then, the cells were subjected to Western blot analysis for recombinant protein expression confirmation and the supernatant (P1) was used to amplify the rBVs to a higher-titer P2 and scaled up to obtain a higher volume of P3 to be titrated (Plate-Forme Technologique Production et Purification de Protéines Recombinantes, Institut Pasteur) and used for protein production.

## Protein expression and purification

His-VP3 polypeptides were produced in the Bac-to-Bac Baculovirus Expression System. Briefly, Sf9 cells (Gibco, Thermo Fisher Scientific) were infected with recombinant Baculoviruses (rBVs) at a multiplicity of infection (MOI) of 5 PFU/cell. Cells were harvested at 96 hr post-infection, washed once with chilled phosphate-buffered saline (PBS), resuspended in lysis buffer 50 mM Tris-HCl (pH 8.0), 500 mM NaCl, 0.1% nonidet P-40 supplemented with protease inhibitors (Complete Mini, Roche), and maintained on ice for 30 min. Thereafter, extracts were sonicated and centrifuged at 13.000 g for 15 min at 4 °C. Supernatants were collected and subjected to immobilized metal-affinity chromatography (IMAC) purification by using a $Ni^{2+}$ affinity column (HiTrap Fast Flow Crude, GE Healthcare). Resin-bound proteins were eluted with elution buffer [50 mM Tris-HCl (pH 8.0), 500 mM NaCl, 500 mM imidazole]. His-tagged VP3 FL, VP3 ΔCt, or VP3 FL $R_{200}D$ -containing fractions were pooled and subjected to size exclusion chromatography using a Superdex 200 column (GE HealthCare). Finally, protein samples were concentrated to a final concentration of ~3 mg/ml by using Centricon filters (Millipore), aliquoted and flash-frozen in liquid nitrogen for –80 °C storage. The His-VP3 FL, His-VP3 ΔCt, and His-VP3 FL $R_{200}D$ fusion proteins obtained from Baculoviruses were used for co-flotation, bio-layer interferometry, and cryo-EM assays.

For His-2xFYVE polypeptide production, *E. coli* BL21 DE3 cells containing the plasmid pET32a(+)–2xFYVE were grown at 37 °C until an OD600 of 0.6 was reached. Protein expression was induced overnight at 18 °C using 1 mM IPTG. Cells were harvested via centrifugation, resuspended in buffer [10 mM TRIS-HCl (pH 8.0), 100 mM NaCl] supplemented with protease inhibitors, and sonicated. The

cell lysate was clarified via centrifugation at 18.000 g for 20 min at 4 °C and the resulting supernatant was subjected to IMAC purification by using a $Ni^{2+}$ affinity column (HiTrap Fast Flow Crude, GE Healthcare). Resin-bound proteins were eluted with elution buffer [10 mM TRIS-HCl (pH 8.0), 100 mM NaCl, 500 mM imidazole]. His-2xFYVE-containing fractions were pooled and subjected to size exclusion chromatography using a Superdex 75 column (GE HealthCare). Finally, protein samples were concentrated to a final concentration of ~3 mg/ml by using Centricon filters (Millipore), aliquoted and flash-frozen in liquid nitrogen for –80 °C storage. The His-2xFYVE was used as a positive control in co-flotation, bio-layer interferometry, and cryo-EM assays.

For GST-2xFYVE polypeptide production, a classical GST fusion protein purification protocol, using pGEX-5X-1 vector (GE Healthcare, 27-4584-01) cloning as described was followed (*Nascimbeni et al., 2017*). Briefly, *E. coli* BL21 DE3 strain maxi-cultures were lysed and subjected to glutathione beads binding for GST purification. The glutathione beads were finally treated with benzamidine-coated beads (GE Healthcare Life Science, 17512301) to remove the factor Xa and dialyzed for the final purification in a 75 mM Kac, 30 mM HEPES, pH 7.4, 5 mM $MgCl_2$ solution. The GST-2xFYVE fusion protein was used as a probe to detect PI3P in immunofluorescence assays.

## Purified proteins quantification and quality control assessment

His-VP3 FL, His-VP3 ΔCt, His-VP3 FL $R_{200}D$, and His-2xFYVE quantification at 280 nm was carried out by recording a full spectrum between 240 and 340 nm. Measurements were done with 60 µL of buffer and sample at 20 °C in a 1 cm quartz cell, reference 105.202-QS.10 (Hellma Analytics, France), using a JASCO V-750 spectrophotometer (JASCO Corporation, Japan). A baseline subtraction at 340 nm was performed with the Spectragryph software to accurately calculate the protein concentration. The assessment of the sample homogeneity was performed by Dynamic Light Scattering (DLS) analysis on a DynaPro Plate Reader III (Wyatt, Santa Barbara, CA, USA) to ensure that the samples did not contain aggregates. A volume of 20 µL of sample was loaded in a 384-well microplate (Corning ref 3540, New-York, USA), with 10 acquisitions of 5 s each at 20 °C, monitored with the DYNAMICS version V7.10.0.21 software (Wyatt, Santa Barbara, CA, USA), three repetitions per measurement. Finally, the protein integrity and purity were assessed by intact mass spectrometry on a Bruker UltrafeXtreme MALDI-TOF/TOF instrument. A volume of 15 µL of protein was passed through a ZipTip C4 and eluted on a MTP 384 ground steel target plate (Bruker-Daltonics, Germany) with 2 µl of a 20 mg/ml α-Cyano-4-hydroxycinnamic acid in 50% acetonitrile, 0.1% trifluoroacetic acid as matrix solution. Data were acquired using Flexcontrol software (Bruker-Daltonics, Germany) and shots were recorded in positive ion linear mode. Mass spectra were externally calibrated in the m/z range of 15–60 kDa with the Protein II (Bruker-Daltonics, Germany) and analyzed with the flexAnalysis software (Bruker).

## Liposome preparation

Liposomes were freshly prepared by the freeze-thaw and extrusion method (F Olson, *Olson et al., 1979*). Briefly, lipids dissolved in chloroform were mixed and placed in glass vials, and the chloroform was evaporated by centrifuging 2 hr at 35 °C in a SpeedVac. The dried lipids were resuspended in a binding buffer composed of 150 mM NaCl, 20 mM Tris-HCl, pH 8 to a final concentration of 5 mM. The preparation was submitted to ten cycles of freezing in liquid nitrogen and thawing in a 37 °C water bath. To generate small unilamellar liposomes, the multilamellar lipids were extruded through polycarbonate membranes (pore size 200 nm, Whatman). The liposomes were used within 1 wk. The lipids POPE (1-palmitoyl-2-oleoyl-sn-glycero-3-phosphoethanolamine), POPC (1-palmitoyl-2-oleoyl-sn-glycero-3-phosphocholine), PI3P [1,2-dioleoyl-sn-glycero-3-phospho'(1'-myo-inosito'–3'-phosphate)], PA (1,2-dioleoyl-sn-glycero-3-phosphate), and PI [1,2-dioleoyl-sn-glycero-3-phospho'(1'-myo-inositol)] were purchased from Avanti Polar Lipids. Liposomes PI3P(-) liposomes were prepared as a mixture of POPE and POPC at a molar ratio of 64:36. Liposomes PI3P(+), PA(+), or PI(+) were prepared as a mixture of POPE and POPC, with either PI3P, PA, or PI at a molar ratio of 64:31:5.

## Liposome co-flotation assay

For liposome co-flotation assays, the following proteins were used: His-VP3 FL, His-VP3 ΔCt, His-VP3 FL $R_{200}D$ produced in the Bac-to-Bac Baculovirus Expression System (Invitrogen), and His-2xFYVE produced using the *Escherichia coli* prokaryotic system. Then, 3 mM liposomes were incubated with 1 µM protein in binding buffer composed of 150 mM NaCl, 20 mM Tris-HCl, pH 8 at 4 °C overnight

(on), in a final volume of 100 µl. Liposomes and liposome-protein complexes were separated from unbound protein by ultracentrifugation of the reaction mixture in 5 mL Optiprep (Axis-Shield PoC AS) density gradients for 1 hr at 40,000×g (SW55Ti Beckman Coulter rotor). The liposome-protein mixture was adjusted to 34% Optiprep concentration in a volume of 300 µl, and loaded onto the bottom of the tube that contained 4.5 ml of 20% Optiprep in 150 mM NaCl, 20 mM Tris-HCl, pH 8, overlaid with 200 µl of the buffer. The three top and the bottom fractions, containing 400 µl of the gradient were used for SDS-PAGE gel analysis and immunoblotting against His-tag and anti-VP3 rabbit polyclonal serum.

## Cryo-electron microscopy

Protein-liposome complexes were prepared as described for co-flotation assays. Then, 4 µl of the samples were spotted on glow-discharged lacey grids (S166-3, EMS) and cryo-fixed by plunge freezing at –180 °C in liquid ethane using a Leica EMGP (Leica, Austria). Grids were observed with a Tecnai F20 electron microscope (Thermo Fisher Scientific). The Tecnai F20 was operated at 200 kV and images were acquired under low-dose conditions using the software EPU (Thermo Fisher Scientific) and a direct detector Falcon II (Thermo Fisher Scientific). For immunogold staining, the purified protein were first incubated with 5 µL of Ni-NTA [nickel (II) nitrilotriacetic acid] gold particles 0.5 mM in a five times gold particles/purified protein proportion in binding buffer to a final volume of 10 µl. This reagent comprises 5 nm gold particles covered with multiple Ni-NTA functionalities incorporated into the ligands on the surface of gold particles. Each $Ni^{2+}$ coordinates with one NTA and two histidines from the His-tagged recombinant protein to form a stable complex with extremely low dissociation constants (*Reddy et al., 2005*). Then, 4 µl of this mix were incubated with 6 µl of 3 mM liposome preparation at 4 °C. Finally, 4 µl of each complex was subjected to cryo-fixation and cryo-EM analyzed as mentioned.

## Bio-layer interferometry (BLI) assays

His-2xFYVE, His-VP3 FL, or His-VP3 ΔCt were immobilized on Octet NTA biosensor (Sartorius) incubated in an Octet^Red 384 system (ForteBio). The functionalized sensors were then washed for 120 s in binding buffer composed of 150 mM NaCl, 20 mM Tris-HCl, pH 8 - BSA (1 mg/mL) before dipping for 600 s into solutions of liposomes diluted in binding buffer - BSA to a final concentration of 5.0, 1.5, or 0.5 mM. Dissociation of the liposome/protein complexes was then monitored in binding buffer - BSA for 300 s. All the experiments were performed in duplicate on two different sensors to account for potential experimental artifacts due to inter-sensor variability. In all cases, potential non-specific interactions were monitored using a sensor coated with the fusion His-Streptavidin as a negative control. The curves were processed using BiaEvaluation software (Biacore).

## Recombinant mammalian expression plasmids

Plasmids encoding EGFP-Rab5 was kindly provided by Philip D. Stahl (Washington University, St. Louis, MO) to our group. Plasmids pcDNA VP3 FL and pcDNA VP3 P2 (P2 all reversed) were constructed by Valli and collaborators in a previous study (*Valli et al., 2012*) and kindly provided by José F. Rodríguez (CSIC, Madrid, Spain) to our group. The plasmids encoding point mutant VP3 proteins ($K_{157}D$, $R_{159}D$, $H_{198}D$, and $R_{200}D$) were obtained by site-directed mutagenesis. For this purpose, the plasmid pcDNA VP3 FL was used as template DNA in polymerase chain reactions (PCR) together with oligonucleotide primer pairs containing changed nucleotides in order to generate the point mutations (*Supplementary file 1*). PCRs were carried out using Pfu DNA polymerase (PB-L, Argentina) and the following conditions: initial DNA denaturation at 95 °C for 5 min followed by 30 cycles of 95 °C for 1 min, 65 °C for 1 min and 72 °C for 6 min and a final extension at 72 °C for 10 min. The size of the amplified products was corroborated by agarose gel electrophoresis. In order to degrade the plasmid mold, the PCR mixtures were incubated with *Dpn*I endonuclease (Thermo Fisher Scientific, USA) at 37 °C for 2 hr followed by heating at 80 °C for 20 min. Aliquots of those reactions were transformed into competent *E. coli* DH10B cells (Invitrogen, USA). Colonies carrying the mutated plasmids were selected in LB plates containing ampicillin (100 µg/mL). Plasmids were purified using the EasyPure Plasmid MiniPrep Kit (TransGen Biotech Co., Ltd, China), and the introduction of mutations was confirmed by nucleotide sequencing (Macrogen service, South Korea). Alignments and prediction of amino acid sequences were performed using the Aligner and Translator tools from the JustBio website (https://justbio.com/).

## Transient transfections and indirect immunofluorescence assay

QM7 cells were grown on coverslips in an M24 multi-well plate for 12 hr to approximately 70% confluence and then plasmids were transfected employing Lipofectamine 3000 (number L300015; Thermo Fisher, Argentina) following the manufacturer's recommendations. After 24 hr p.i., the cells were fixed with 4% paraformaldehyde (PFA) for 15 min at room temperature (RT), washed with PBS (pH 7.4), and permeabilized with 0.05% saponin in PBS containing 0.2% BSA for 20 min at RT. Then the cells were incubated with anti-VP3 primary antibodies overnight (ON) at 4 °C in a humidity chamber, and after extensive washing, cells were incubated with secondary antibodies conjugated with Alexa Fluor Cy3 for 1 hr 30 min, followed by extensive washes in PBS. The coverslips were mounted in Mowiol plus Hoechst and analyzed by Confocal Laser Scanning Microscopy (CLSM). Images were captured using an Olympus FluoView TM FV1000 confocal microscope (Olympus, Argentina) with FV10-ASW (version 01.07.00.16) software and processed using the ImageJ software (*Schindelin et al., 2012*). For quantification of VP3 cellular distribution (*Figure 2D*), the percentage of cells with vesicular distribution of the red signal was calculated out of approximately 30 cells per construct and experiment. For quantification of the co-localization of VP3 and either EGFP-Rab5 or EGFP-2xFYVE, the Manders $M_2$ coefficient was calculated out of approximately 30 cells per construct and experiment (*Figure 2—figure supplement 1*, *Figure 3A*, respectively). The M2 coefficient, which reflects co-localization of signals, is defined as the ratio of the total intensities of magenta image pixels for which the intensity in the blue channel is above zero to the total intensity in the magenta channel. JACoP plugin was utilized to determine M2 (*Bolte and Cordelières, 2006*). For VP3 puncta co-distributing with EEA1 and GST-2xFYVE the number of puncta co-distributing for the three signals was manually determined out of approximately 40 cells per construct and experiment per 200 μm² (*Figure 3B*).

## Reverse genetics

For the generation of recombinant IBDV, we used a modification of the reverse genetics system described by Qi and coworkers (*Qi et al., 2007*), in which the full-length sequences of the IBDV segments, flanked by a hammerhead ribozyme at the 5'-end and a hepatitis delta ribozyme at the 3'-end, are expressed under the control of an RNA polymerase II promoter. For its construction, the complete segments of IBDV Soroa were amplified from viral dsRNA using a sequence-independent amplification method (*Potgieter et al., 2009*) and cloned in plasmid pJET1.2 (Thermo Fisher Scientific). The hammerhead ribozymes, specific for each segment, are based in the HamRz-R, described by Yanay and cols. (*Yanai et al., 2006*), with 10 nucleotides of sequence complementary to each IBDV segment stabilizing Stem I. At the 3'-end, we used the sequence of the antigenomic hepatitis delta ribozyme HDVagrz SC (*Ghanem et al., 2012*). DNA fragments containing the ribozyme sequences were constructed by PCR-based gene synthesis with oligonucleotides designed by the Assembly PCR Oligo Maker software (*Rydzanicz et al., 2005*). Each PCR-amplified IBDV full-length segment were cloned, flanked by the corresponding hammerhead ribozyme and HDVagrz SC, between the *Eco*RI and *Not*I sites of plasmid pCAGEN a gift from Connie Cepko; Addgene plasmid # 11160; (*Matsuda and Cepko, 2004* by Gibson assembly *Gibson et al., 2009*). The correctness of the sequences in both plasmids, pCAGEN.Hmz.SegA.Hdz (SegA) and pCAGEN.Hmz.SegB.Hdz (SegB), was assessed by sequencing (Eurofins Scientific). The plasmid pCAGEN.Hmz.SegA.$R_{200}$D.Hdz (SegA.$R_{200}$D) was purchased to GenScript and the mutation was verified by sequencing of the complete plasmid (Eurofins Scientific).

## Recombinant viruses foci forming units assay

QM7 cells were grown in M24 multi-well plates for 12 hr to approximately 90–95% confluency and then 800 ng (400 ng of plasmid with each segment) were transfected by using Lipofectamine 2000 (number 11668027; Thermo Fisher, Spain) following the manufacturer's recommendations. When performing single segment transfections (SegA, SegB, and SegA.$R_{200}$D), 400 ng of the plasmid pCAGIG a gift from Connie Cepko; Addgene plasmid #11159; *Matsuda and Cepko, 2004* was added to 400 ng of those containing the viral segments to reach a total of 800 ng of total DNA per transfection. pCAGIG is a mammalian expression vector that shares the backbone with the pCAGEN vector used for the expression of the viral segments. The following transfections were performed in triplicate: SegA, SegB, SegA.$R_{200}$D, SegA + SegB, or SegA.$R_{200}$D+SegB. At 8 hr post-transfection (p.t.) the supernatants were discarded and the monolayers were recovered for further plating on M6 multi-well plates

containing non transfected QM7 cells. For monolayers transfected with SegA, SegB, SegA.R$_{200}$D, and SegA.R$_{200}$D+SegB half of the transfected monolayer was plated on fresh cells. For SegA +SegB, which leads to the production of wild-type virus, several dilutions were tested, always in triplicate. To limit virus diffusion and facilitating the foci forming Avicel RC-591 (FMC Biopolymer) was added to the M6 multi-well plates. 72 hr p.i., the monolayers were 16% formaldehyde-fixated and stained with Coomassie R250 (BioRad).

## Statistical analysis

One-way ANOVA followed by a Tukey's HSD (honestly significant difference) test were performed using AnalystSoft Inc, StatPlus:mac - Version v8 (*AnalystSoft Inc, 2025*). Plotting was done using the *ggplot2* package *Wickham, 2016* in R, with assistance from the RStudio software (*RStudio Team, 2020*).

## Bioinformatics analysis

Multiple sequence alignment and the percent identity matrix were performed with Clustal OMEGA (v1.2.4) (*Sievers et al., 2011*) implemented at EMBL's European Bioinformatics Institute ('EMBL's European Bioinformatics Institute,' n.d.). Accession numbers of the sequences: Infectious bursal disease virus (AF_140705.1); Blotched snakehead virus (YP_052872.1); Victorian trout aquabirnavirus (YP_009255397.1); Tellina virus 2 (YP_010084301.1); Yellowtail ascites virus (NP_690805.1); Infectious pancreatic necrosis virus (NP_047196.1); Tasmanian aquabirnavirus (YP_009177608.1); Lates calcarifer birnavirus (YP_010086267.1); and Tellina virus 1 (YP_009509080.1). Alignment visualization was done with the R *ggmsa* package (*Zhou et al., 2022*) with the assistance from the RStudio software (*RStudio Team, 2020*). Amino acids are colored according to their side-chain chemistry. Protein sequence logos annotation is displayed on top of the amino acid alignment.

The model of VP3 FL from IBDV [strain Soroa, GenBank AF140705.1, residues 756–1012 (257 residues) from the polyprotein] was generated using a local copy of AlphaFold-2 *Jumper et al., 2021*, installed using the open-source code available at https://github.com/deepmind/alphafold (*Zidek and Tomlinson, 2021*; accessed on 1 September 2021). Runs were performed on a Centos 7 workstation with 64 CPUs and 4 GeForce RTX 2080 Ti GPUs, using the casp14 preset and including all PDB templates present in the database. The program produces a per-residue confidence metric termed pLDTT on a scale from 0 to 100. pLDTT values higher than 70 reflect reliable models with correct backbone predictions, and those with values higher than 90 correspond to models with both reliable backbone and side-chain orientation predictions. pLDTT values lower than 50 are a strong predictor of disorder, and regions with such values are either unstructured under physiological conditions or only structured as part of a complex.

## Reagents and antibodies

Western blot and confocal laser scanning microscopy (CLSM) analyses were carried out using rabbit anti-VP3 specific sera followed by horseradish peroxidase (HRP)-conjugated anti-rabbit secondary antibodies (number A0545), purchased from Sigma-Aldrich (Buenos Aires, Argentina). For early endosomes detection by CLSM, we used anti-EEA1 mouse monoclonal antibody (1:200 dilution; BD Transduction Laboratories, 610456), and for anti-GST detection, we used Alexa Fluor 649-labeled goat anti-GST (Rockland-Inc, 600-143-200). Fluorescently labeled secondary antibodies, all diluted 1:200 from Invitrogen, included Alexa Fluor 546 donkey anti-mouse (A10036), and Alexa Fluor 546 donkey anti-rabbit (A10040).

## Far-UV circular dichroism (CD) of VP3 proteins

His-VP3 FL and His-VP3 FL R$_{200}$D at a concentration of 2 mg/ml in 50 mM Tris-HCl pH 8, 500 mM NaCl were analyzed by far-UV CD (180–260 nm). CD spectra were obtained with an AVIV CD spectropolarimeter model 215 using a 0.02 cm path length cell at RT. Five successive scans were averaged and the background spectrum of the sample buffer, acquired under identical conditions, was subtracted. The resulting corrected CD intensities were then converted to Δε per residue. Secondary structure contents were estimated from the far-UV CD spectra using the CDSSTR routine (*Johnson, 1999*) of the DICHROWEB server (*Whitmore and Wallace, 2008*; *Whitmore and Wallace, 2004*) run on the SP175 reference dataset (*Lees et al., 2006*), containing 72 proteins representing a large panel

of secondary structures. Similar results were obtained on different datasets (*Sreerama and Woody, 2000*) or by using the CONTIN/LL routine (*Provencher and Glöckner, 1981*).

## Coarse grain (CG) models of VP3 constructs

Three main protein constructs derived from VP3 were considered: 1. VP3 Ct. A peptide comprising the last 36 residues of VP3 (223–257). VP3 Ct has an overall electrical charge of +5, and is absent from the available crystal structure of VP3. Consequently, it was assumed to be a disordered domain and was modeled as unstructured. 2. VP3 ΔNt. A shorter version of VP3 that lacks the first 81 residues leading to the Nt. It consists of the crystallized structure of the protein core (residues 82–222, available from the Protein Data Bank PDB ID 2R18), concatenated with the VP3 Ct peptide. As the first nine residues of the protein core are also missing from resolved structure, they were modeled as unstructured, similar to the VP3 Ct fragment. VP3 ΔNt was selected primarily because it contains the structural information of the crystallized protein, which provides valuable insights into the protein's overall structure and function. In addition, the construct includes the Ct peptide (VP3 Ct), which is crucial for membrane binding. Although the Nt 81 residues are absent in this construct, their exclusion was found to be inconsequential in terms of evaluating the protein's binding energy and positioning on the membrane surface (compare, for example, VP3 ΔNt and VP3 FL in *Figure 5*). 3. VP3 FL. The full-length VP3 protein, which was obtained as an AlphaFold2 prediction (*Jumper et al., 2021*). The structure of the protein predicted by AlphaFold2 not only provided insight into the overall architecture of VP3, but it also reinforced our model of unstructured VP3 Ct. It is worth noting that the Nt 81 residues, which are absent in VP3 ΔNt, are predicted to be in a modular conformation, constituting a globular domain separated by a loop from the globular domain of the core region (crystallized residues 82–222). The structural information provided by the full-length VP3 model is, therefore, useful in understanding the interaction with lipid membranes. In addition to VP3 Ct, VP3 ΔNt, and VP3 FL, two mutants of VP3 ΔNt were simulated: VP3 ΔNt $R_{200}$D, in which $R_{200}$ was replaced by aspartate (D), and VP3 ΔNt P2 Mut, where the whole P2 region ($R_{200}$, $H_{198}$, $R_{159}$, and $K_{157}$) was replaced by D. All proteins were modeled with the MARTINI force-field (*de Jong et al., 2013*) [whenever needed, the secondary structure of the non-disordered protein domains was restrained by means of the ElNeDyn elastic network model (elnedyn22)] (*Periole et al., 2009*; *Poma et al., 2017*). In this case, harmonic potentials with a force constant of 1000 kJ mol$^{-1}$ nm$^2$ were applied between backbone particles separated by less than 1.4 nm. In the construction of VP3 ΔNt, elastic network restrictions were applied to residues 91–220, which was the fragment resolved from by crystallography. However, in VP3 FL, the elastic network was applied to all globular domains, while leaving the linkers and the last 36 residues at the Ct unrestricted.

## Molecular theory calculations

The molecular theory (MT) used to assess protein adsorption is detailed in references (*Chiarpotti et al., 2021*; *Ramírez et al., 2019*). Briefly, this mean-field method yields the distribution of macromolecules in an anisotropic system, through the minimization of a model free-energy functional that accounts for electrostatic and steric interactions, the configurational and translational entropy of the macromolecules, the translational entropy of the free species (protons, hydroxyls, and salt ions) and the acid-base equilibrium of all ionizable molecules in the system. In the present case, the membrane was modeled as a dielectric slab of thickness h, with a dielectric constant $\varepsilon_M$, which is in contact with a solution of dielectric constant $\varepsilon_S$. Far from the membrane (z→∞), the solution is in contact with an infinite bulk that contains the proteins (1 µM concentration), all the free species, and a fixed pH = 8. The rigid membrane contained 15% of ionizable groups on its surface, in order to mimic the 5% of ternary ionizable PI3P. One-third of these groups were assigned a p$K_a$ of 2.5, corresponding to the phosphate group of the glycerol moiety, while the p$K_a$ for the rest was set to 6.5, corresponding to the phosphoesther groups of inositol. MT requires the input of a large set of protein conformations, which are subsequently reweighted by the theory. These conformations were generated by Molecular Dynamics simulations of the proteins in water. VP3 Ct, VP3 ΔNt, VP3 FL, VP3 ΔNt $R_{200}$D, and VP3 ΔNt P2 Mut, were modeled using MARTINI-2.2 parameters (*Marrink et al., 2007*). The proteins were hydrated in MARTINI polarizable water. As stated before, the proteins globular segments (residues 92–222 in VP3 FL, VP3 ΔNt and its mutants, and residues 7–79 additionally in VP3 FL) were restricted through the application of the elastic network ElNeDyn (*Marrink et al., 2007*) with a spring constant

of 1000 kJ.mol⁻¹ applied to backbone beads (BB) within a cutoff distance of 1.4 nm. Each protein was simulated for 100 ns, after equilibration, in the NPT ensemble using GROMACS 2021.3. A cluster analysis was performed in order to obtain the most representative conformations of the simulated ensemble. This method groups conformations by structural similarity, using the root mean squared deviation (RMSD) of the coordinates calculated for all conformer pairs along the trajectory, clustering structures with RMSD ≤0.3 nm. This procedure leads to 50 conformers for each protein construct. Each conformer was subsequently rotated 30 times randomly to produce a set of 1500 configurations for the MT calculations.

## Molecular dynamics simulations of VP3 on a lipid bilayer

MARTINI-2.2 was also used to model the lipid molecules in our MD simulations. The bilayer was assembled using an insane utility (*Wassenaar et al., 2015*), while hydration with polarizable water molecules and ions was performed with the solvate utility of the GROMACS package. The bilayer had a symmetric lipid composition of 30% 1,2-dioleoyl-sn-glycero-3-phosphocholine (DOPC), 65% of 1,2-dioleoyl-sn-glycero-3-phosphoethanolamine (DOPE), and 5% 1,2-dioleoyl-sn-glycero-3-phosphatidylinositol-3-phosphate (PI3P), leading to a lipid composition of 111:231:18 (DOPC, DOPE, and PI3P) in each leaflet. The protein was placed at a distance of 10 nm from the membrane surface, to avoid a possible initial binding bias. The system was then equilibrated and finally simulated for 500 ns. Both equilibration and production were in the NPT ensemble. A semi-isotropic Parinello-Rahman barostat (*Parrinello and Rahman, 1981*) with a 12 ps time constant, and lateral and normal compressibilities of $3\times10^{-4}$ bar⁻¹ were used to keep the pressure at 1.0 bar. The v-rescale thermostat, with a time constant of 1 ps, was used to keep the temperature at 325 K. In all cases, the integration timestep was 20 fs. Long range electrostatics were calculated using the particle mesh Ewald (PME) method (*Darden et al., 1993*) with a real-space cut-off of 1.2 nm.

## Acknowledgements

This project was partially supported by: the National Agency for Scientific, and Technological Promotion, Ministry of Science, Technology and Innovation, through grants PICT 2016–0528 and 2019–01324 to LRD; PICT 2015–2210 to FAZ; PICT 2019–01889 to LMP, and PICT 2020–3795 to MGDP; the National Scientific and Technical Research Council through grants PIP 2015–2017 11220150100114CO and 2021–2023 11220200103139CO to LRD; 2021–2023 11220200103195CO to LMP and 11220200103223CO to MGDP; and by the National University of Cuyo through grants 2013–2015 M006, 2016–2018 M029, 2019–2021 M071 and 2022–2024 M012 to LRD For stays performed by LRD in the laboratory of Structural Virology at the Institut Pasteur in Paris, we are grateful for the support of National University of Cuyo through the Teaching Staff Mobility Program in 2018–2019, the Ministry of Education through the Program scholarships for training abroad in Science and Technology (BEC.AR 2018), and the International Union of Biochemistry and Molecular Biology (IUBMB) through a '2020 Mid-Career Research Fellowship.' The work in Madrid was supported by grants from the Spanish Ministry of Science and Innovation (PID2020-113287RB-I00) and the Comunidad Autónoma de Madrid (P2018/NMT-4389) to JRC. At the IHEM, we sincerely appreciate Elisa Bocanegra, Norberto Domizio, and Jorge Ibañez for valuable technical assistance in CLSM handling. At the Institut Pasteur in Paris, we are grateful to Gérard Pehau-Arnaudet and the staff at the Ultrastructural BioImaging core facility for image acquisition and analysis; to Patrick England and Sébastien Brûlé and the staff of the Molecular Biophysics platform for purified protein analyses, CD and BLI assays, to Stéphane Petres and members at the Production and Purification of Recombinant Proteins Technological Platform, and to the staff of the Proteomics platform.

## Additional information

### Competing interests

Laura Ruth Delgui: Reviewing editor, eLife. The other authors declare that no competing interests exist.

## Funding

| Funder | Grant reference number | Author |
| --- | --- | --- |
| Ministry of Science, Technology and Innovation | PICT 2016-0528 | Laura Ruth Delgui |
| Ministry of Science, Technology and Innovation | PICT 2019-01324 | Laura Ruth Delgui |
| Ministry of Science, Technology and Innovation | PICT 2015-2210 | Flavia A Zanetti |
| Ministry of Science, Technology and Innovation | PICT 2019-01889 | Luis Mariano Polo |
| Ministry of Science, Technology and Innovation | PICT 2020-3795 | Mario Del Pópolo |
| National Scientific and Technical Research Council | PIP 2015-2017 11220150100114CO | Laura Ruth Delgui |
| National Scientific and Technical Research Council | PIP 2021-2023 11220200103195CO | Laura Ruth Delgui |
| National Scientific and Technical Research Council | PIP 11220200103223CO | Mario Del Pópolo |
| National Scientific and Technical Research Council | 2021–2023 11220200103139CO | Laura Ruth Delgui |
| National University of Cuyo | 2013-2015 M006 | Laura Ruth Delgui |
| National University of Cuyo | 2016-2018 M029 | Laura Ruth Delgui |
| National University of Cuyo | 2019-2021 M071 | Laura Ruth Delgui |
| National University of Cuyo | 2022-2024 M012 | Laura Ruth Delgui |
| National University of Cuyo | Teaching Staff Mobility Program in 2018-2019 | Laura Ruth Delgui |
| Ministry of Education | BEC.AR 2018 | Laura Ruth Delgui |
| International Union of Biochemistry and Molecular Biology | 2020 Mid-Career Research Fellowship | Laura Ruth Delgui |
| Spanish Ministry of Science and Innovation | PID2020-113287RB-I00 | Jose R Castón |
| Comunidad Autónoma de Madrid | P2018/NMT-4389 | Jose R Castón |

The funders had no role in study design, data collection and interpretation, or the decision to submit the work for publication.

### Author contributions

Flavia A Zanetti, Pablo Guardado-Calvo, Andres Ferrino-Iriarte, Sarah Dubois, Etienne Morel, Victoria Alfonso, Milton Osmar Aguilera, María E Celayes, Luis Mariano Polo, Laila Suhaiman, Vanesa V Galassi, Maria V Chiarpotti, Carolina Allende-Ballestero, Javier M Rodriguez, Oscar Taboga, Investigation; Ignacio Fernandez, Conceptualization, Formal analysis; Eduard Baquero, Conceptualization; Jose R Castón, Resources; Diego Lijavetzky, Data curation, Investigation; María I Colombo, Resources, Writing – review and editing; Mario Del Pópolo, Conceptualization, Data curation, Writing – original draft, Writing – review and editing; Félix A Rey, Data curation, Writing – review and editing; Laura Ruth Delgui, Conceptualization, Data curation, Formal analysis, Supervision, Funding acquisition, Investigation, Methodology, Writing – original draft, Writing – review and editing

### Author ORCIDs

Sarah Dubois ⓘ http://orcid.org/0009-0000-7504-7135
Etienne Morel ⓘ http://orcid.org/0000-0002-4763-4954
Javier M Rodriguez ⓘ https://orcid.org/0000-0003-0146-9903
Jose R Castón ⓘ https://orcid.org/0000-0003-2350-9048

Mario Del Pópolo [iD] https://orcid.org/0000-0002-1435-2424
Félix A Rey [iD] https://orcid.org/0000-0002-9953-7988
Laura Ruth Delgui [iD] https://orcid.org/0000-0002-3647-3593

Reviewer #1 (Public review): https://doi.org/10.7554/eLife.97261.3.sa1
Reviewer #3 (Public review): https://doi.org/10.7554/eLife.97261.3.sa2
Author response https://doi.org/10.7554/eLife.97261.3.sa3

## Additional files

### Supplementary files

Supplementary file 1. Primers. All the primers used for this work. * For the first four-point mutants, the nucleotides that allow to introduce amino acid changes in the VP3 protein are indicated in italic bold letters. The underlined nucleotides indicate restriction sites.

MDAR checklist

### Data availability

All data generated or analysed during this study are included in the manuscript and supporting files.

The following previously published datasets were used:

| Author(s) | Year | Dataset title | Dataset URL | Database and Identifier |
|---|---|---|---|---|
| Da Costa B, Soignier S, Chevalier C, Henry C, Thory C, Huet JC, Delmas B | 2018 | Blotched snakehead virus is a new aquatic birnavirus that is slightly more related to avibirnavirus than to aquabirnavirus | https://www.ncbi.nlm.nih.gov/protein/YP_052872.1 | NCBI Protein, YP_052872.1 |
| Mohr PG, Moody NJ, Williams LM, Hoad J, Crane MS | 2018 | Molecular characterization of Tasmanian aquabirnaviruses from 1998 to 2013 | https://www.ncbi.nlm.nih.gov/protein/YP_009255397.1 | NCBI Protein, YP_009255397.1 |
| Blake S, Ma JY, Caporale DA, Jairath S, Nicholson BL | 2021 | Phylogenetic relationships of aquatic birnaviruses based on deduced amino acid sequences of genome segment A cDNA | https://www.ncbi.nlm.nih.gov/protein/YP_010084301.1 | NCBI Protein, YP_010084301.1 |
| Suzuki S, Kimura M, Kusuda R | 2018 | The complete nucleotide sequence of the polyprotein and VP5 gene of a marine birnavirus | https://www.ncbi.nlm.nih.gov/protein/NP_690805.1 | NCBI Protein, NP_690805.1 |
| Duncan R, Dobos P | 2018 | The nucleotide sequence of infectious pancreatic necrosis virus (IPNV) dsRNA segment A reveals one large ORF encoding a precursor polyprotein | https://www.ncbi.nlm.nih.gov/protein/NP_047196.1 | NCBI Protein, NP_047196.1 |
| Mohr PG, Moody NJ, Williams LM, Hoad J, Crane MS | 2018 | Molecular characterization of Tasmanian aquabirnaviruses from 1998 to 2013 | https://www.ncbi.nlm.nih.gov/protein/YP_009177608.1 | NCBI Protein, YP_009177608.1 |
| Nobiron I, Galloux M, Henry C, Torhy C, Boudinot P, Lejal N, Da Costa B, Delmas B | 2018 | Genome and polypeptides characterization of Tellina virus 1 reveals a fifth genetic cluster in the Birnaviridae family | https://www.ncbi.nlm.nih.gov/protein/YP_009509080.1 | NCBI Protein, YP_009509080 |
| Chen J, Toh X, Ong J, Wang Y, Teo X, Lee B, Wong P, Chee S, Wee D, Wang A, Ng Y, Tan BM, Khor D, Chong S | 2022 | Detection and characterisation of a novel marine Birnavirus isolated from Asian Seabass in Singapore | https://www.ncbi.nlm.nih.gov/protein/YP_010086267.1 | NCBI Protein, YP_010086267.1 |

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
