## [Editor Report · eLife Assessment]

Zanetti et al use **convincing** biophysical and cellular assays to investigate the interaction of the birnavirus VP3 protein with the early endosome lipid PI3P. The study provides **valuable** insights and will be of interest to virologists. In future studies, it would be interesting to demonstrate that VP3-PIP3P is a specific interaction and not a general interaction with other PIPs.

---

## [Referee Report · Reviewer #1 (Public review)]

Summary:

Zanetti et al use biophysical and cellular assays to investigate the interaction of the birnavirus VP3 protein with the early endosome lipid PI3P. The major novel finding is that association of the VP3 protein with an anionic lipid (PI3P) appears to be important for viral replication, as evidenced through a cellular assay on FFUs.

Strengths:

Support previously published claims that VP3 associates with early endosome membrane, potentially through binding to PI3P. The finding that mutating a single residue (R200) critically affects early endosome binding and that the same mutation also inhibits viral replication suggests a very important role for this binding in the viral life cycle.

Weaknesses:

The manuscript is relatively narrowly focused: the specifics of the bi-molecular interaction between the VP3 of an unusual avian virus and a host cell lipid (PIP3). Further, the affinity of this interaction is low and its specificity relative to other PIPs is not tested, leading to questions about whether VP3-PI3P binding is relevant.

---

## [Referee Report · Reviewer #3 (Public review)]

Summary:

infectious bursal disease virus (IBDV) is a birnavirus and an important avian pathogen. Interestingly, IBDV appears to be a unique dsRNA virus that uses early endosomes for RNA replication that is more common for +ssRNA viruses such as for example SARS-CoV-2.

This work builds on previous studies showing that IBDV VP3 interacts with PIP3 during virus replication. The authors provide further biophysical evidence for the interaction and map the interacting domain on VP3.

Strengths:

Detailed characterization of the interaction between VP3 and PIP3 identified R200D mutation as critical for the interaction. Cryo-EM data show that VP3 leads to membrane deformation.

Comments on revisions:

I have no further comments. The authors have addressed my questions and concerns. I congratulate the authors on their work!

---

## [Author Response]

The following is the authors’ response to the current reviews.

**Public Reviews:**

**Reviewer #1 (Public review):**
Summary:Zanetti et al use biophysical and cellular assays to investigate the interaction of the birnavirus VP3 protein with the early endosome lipid PI3P. The major novel finding is that association of the VP3 protein with an anionic lipid (PI3P) appears to be important for viral replication, as evidenced through a cellular assay on FFUs.Strengths:Support previously published claims that VP3 associates with early endosome membrane, potentially through binding to PI3P. The finding that mutating a single residue (R200) critically affects early endosome binding and that the same mutation also inhibits viral replication suggests a very important role for this binding in the viral life cycle.Weaknesses:The manuscript is relatively narrowly focused: the specifics of the bi-molecular interaction between the VP3 of an unusual avian virus and a host cell lipid (PIP3). Further, the affinity of this interaction is low and its specificity relative to other PIPs is not tested, leading to questions about whether VP3-PI3P binding is relevant.

Regarding the manuscript’s focus, we challenge the notion that studying a single bi-molecular interaction makes the scope of the paper overly narrow. This interaction—between VP3 and PI3P—plays a critical role in the replication of the birnavirus, which is the central theme of our work. Moreover, identifying and understanding such distinct interactions is a fundamental aspect of molecular virology, as they shed light on the precise mechanisms that viruses exploit to hijack the host cell machinery. Consequently, far from being narrowly focused, we believe our work contributes to the broader understanding of host-pathogen interactions.

As for the low affinity of the VP3-PI3P interaction, we argue that this is not a limitation but rather a biologically relevant feature. As discussed in the manuscript, the moderate strength of this interaction is likely critical for regulating the turnover rate of VP3/endosomal PI3P complexes, which in turn could optimize viral replication efficiency. A stronger affinity might trap VP3 on the endosomal membrane, whereas weaker interactions might reduce its ability to efficiently target PI3P. Thus, the observed affinity may reflect a fine-tuned balance that supports the viral life cycle.

With regard to specificity, we emphasize that in the context of the paper, we refer to biological specificity, which is not necessarily the same as chemical specificity. The binding of PI3P to early endosomes is “biologically” preconditioned by the distribution of PI3P within the cell. PI3P is predominantly localized in endosomal membranes, which “biologically precludes” interference from other PIPs due to their distinct cellular distributions. Moreover, while early endosomes also contain other anionic lipids, our work demonstrates that among these, PI3P plays a distinctive role in VP3 binding. This highlights its functional relevance in the context of early endosome dynamics.

**Reviewer #3 (Public review):**
Summary:Infectious bursal disease virus (IBDV) is a birnavirus and an important avian pathogen. Interestingly, IBDV appears to be a unique dsRNA virus that uses early endosomes for RNA replication that is more common for +ssRNA viruses such as for example SARS-CoV-2. This work builds on previous studies showing that IBDV VP3 interacts with PIP3 during virus replication. The authors provide further biophysical evidence for the interaction and map the interacting domain on VP3.Strengths:Detailed characterization of the interaction between VP3 and PIP3 identified R200D mutation as critical for the interaction. Cryo-EM data show that VP3 leads to membrane deformation.

We thank the reviewer for the feedback.

The following is the authors’ response to the original reviews.

**Public Reviews:**

**Reviewer #1 (Public Review):**
Summary:Zanetti *et al*. use biophysical and cellular assays to investigate the interaction of the birnavirus VP3 protein with the early endosome lipid PI3P. The major novel finding is that the association of the VP3 protein with an anionic lipid (PI3P) appears to be important for viral replication, as evidenced through a cellular assay on FFUs.Strengths:Supports previously published claims that VP3 may associate with early endosomes and bind to PI3P-containing membranes. The claim that mutating a single residue (R_200_) critically affects early endosome binding and that the same mutation also inhibits viral replication suggests a very important role for this binding in the viral life cycle.Weaknesses:The manuscript is relatively narrowly focused: one bimolecular interaction between a host cell lipid and one protein of an unusual avian virus (VP3-PI3P). Aspects of this interaction have been described previously. Additional data would strengthen claims about the specificity and some technical issues should be addressed. Many of the core claims would benefit from additional experimental support to improve consistency.

Indeed, our group has previously described aspects of the VP3-PI3P interaction, as indicated in lines 100-105 from the manuscript. In this manuscript, however, we present biochemical and biophysical details that have not been reported before about how VP3 connects with early endosomes, showing that it interacts directly with the PI3P. Additionally, we have now identified a critical residue in VP3—the R_200_—for binding to PI3P and its key role in the viral life cycle. Furthermore, the molecular dynamics simulations helped us come up with a mechanism for VP3 to connect with PI3P in early endosomes. This constitutes a big step forward in our understanding of how these "*non-canonical*" viruses replicate.

We have now incorporated new experimental and simulation data; and have carefully revised the manuscript in accordance with the reviewers’ recommendations. We are confident that these improvements have further strengthened the manuscript.

**Reviewer #2 (Public Review):**
Summary:Birnavirus replication factories form alongside early endosomes (EEs) in the host cell cytoplasm. Previous work from the Delgui lab has shown that the VP3 protein of the birnavirus strain infectious bursal disease virus (IBDV) interacts with phosphatidylinositol-3-phosphate (PI3P) within the EE membrane (Gimenez et al., 2018, 2020). Here, Zanetti *et al*. extend this previous work by biochemically mapping the specific determinants within IBDV VP3 that are required for PI3P binding in vitro, and they employ in silico simulations to propose a biophysical model for VP3-PI3P interactions.Strengths:The manuscript is generally well-written, and much of the data is rigorous and solid. The results provide deep knowledge into how birnaviruses might nucleate factories in association with EEs. The combination of approaches (biochemical, imaging, and computational) employed to investigate VP3-PI3P interactions is deemed a strength.Weaknesses:(1) Concerns about the sources, sizes, and amounts of recombinant proteins used for co-flotation: Figures 1A, 1B, 1G, and 4A show the results of co-flotation experiments in which recombinant proteins (control His-FYVE v. either full length or mutant His VP3) were either found to be associated with membranes (top) or non-associated (bottom). However, in some experiments, the total amounts of protein in the top + bottom fractions do not appear to be consistent in control v. experimental conditions. For instance, the Figure 4A western blot of His-2xFYVE following co-flotation with PI3P+ membranes shows almost no detectable protein in either top or bottom fractions.

Liposome-based methods, such as the co-flotation assay, are well-established and widely regarded as the preferred approach for studying protein-phosphoinositide interactions. However, this approach is rather qualitative, as density gradient separation reveals whether the protein is located in the top fractions (bound to liposomes) or the bottom fractions (unbound). Our quantifications aim to demonstrate differences in the bound fraction between liposome populations with and without PI3P. Given the setting of the co-flotation assays, each protein-liposome system [2xFYVE-PI3P(-), 2xFYVE-PI3P(+), VP3-PI3P(-), or VP3-PI3P(+)] is assessed separately, and even if the experimental conditions are homogeneous, it is not surprising to observe differences in the protein level between different experiments. Indeed, the revised version of the manuscript includes membranes with more similar band intensities, as depicted in the new versions of Figures 1 and 4.

Reading the paper, it was difficult to understand which source of protein was used for each experiment (i.e., *E. coli* or baculovirus-expressed), and this information is contradicted in several places (see lines 358-359 v. 383-384). Also, both the control protein and the His-VP3-FL proteins show up as several bands in the western blots, but they don't appear to be consistent with the sizes of the proteins stated on lines 383-384. For example, line 383 states that His-VP3-FL is ~43 kDa, but the blots show triplet bands that are all below the 35 kDa marker (Figures 1B and 1G). Mass spectrometry information is shown in the supplemental data (describing the different bands for His-VP3-FL) but this is not mentioned in the actual manuscript, causing confusion. Finally, the results appear to differ throughout the paper (see Figures 1B v. 1G and 1A v. 4A).

Thank you for pointing out these potentially confusing points in the previous version of the manuscript. Indeed, we were able to produce recombinant VP3 from the two sources: Baculovirus and *Escherichia coli*. Initially, we opted for the baculovirus system, based on evidence from previous studies showing that it was suitable for ectopic expression of VP3. Subsequently, we successfully produced VP3 using *Escherichia coli*. On the other side, the fusion proteins His-2xFYVE and GST-2xFYVE were only produced in the prokaryotic system, also following previous reported evidence. We confirmed that VP3, produced in either system, exhibited similar behavior in our co-flotation and bio-layer interferometry (BLI) assays. However, the results of co-flotation and BLI assays shown in Figs. 1 and 4 were performed using the His-VP3 FL, His-VP3 FL R_200_D and His-VP3 FL DCt fusion proteins produced from the corresponding baculoviruses. We have clarified this in the revised version of our manuscript. Please, see lines 430-432.

Additionally, we have made clear that the His-VP3 FL protein purification yielded four distinct bands, and we confirmed their VP3 identity through mass spectrometry in the revised version of the manuscript. Please, see lines 123-124.

Finally, we replaced membranes for Figs. 4A and 1G (left panel) with those with more similar band intensities. Please, see the new version of Figures 1 and 4.

(2) Possible "other" effects of the R_200_D mutation on the VP3 protein. The authors performed mutagenesis to identify which residues within patch 2 on VP3 are important for association with PI3P. They found that a VP3 mutant with an engineered R_200_D change (i) did not associate with PI3P membranes in co-floatation assays, and (ii) did not co-localize with EE markers in transfected cells. Moreover, this mutation resulted in the loss of IBDV viability in reverse genetics studies. The authors interpret these results to indicate that this residue is important for "mediating VP3-PI3P interaction" (line 211) and that this interaction is essential for viral replication. However, it seems possible that this mutation abrogated other aspects of VP3 function (e.g., dimerization or other protein/RNA interactions) aside from or in addition to PI3P binding. Such possibilities are not mentioned by the authors.

The arginine amino acid at position 200 of VP3 is not located in any of the protein regions associated with its other known functions: VP3 has a dimerization domain located in the second helical domain, where different amino acids across the three helices form a total of 81 interprotomeric close contacts; however, R_200_ is not involved in these contacts (Structure. 2008 Jan;16(1):29-37, doi:10.1016/j.str.2007.10.023); VP3 has an oligomerization domain mapped within the 42 C-terminal residues of the polypeptide, i.e., the segment of the protein composed by the residues at positions 216-257 (*J Virol*. 2003 Jun;77(11):6438–6449, doi: 10.1128/jvi.77.11.6438-6449.2003); VP3’s ability to bind RNA is facilitated by a region of positively-charged amino acids, identified as P1, which includes K_99_, R_102_, K_105_, and K_106_ (*PLoS One*. 2012;7(9):e45957, doi: 10.1371/journal.pone.0045957). Furthermore, our findings indicate that the R_200_D mutant retains a folding pattern similar to the wild-type protein, as shown in Figure 4B. All these lead us to conclude that the loss of replication capacity of R_200_D viruses results from impaired, or even loss of, VP3-PI3P interaction.

We agree with the reviewer that this is an important point and have accordingly addressed it in the Discussion section of the revised manuscript. Please, see lines 333-346.

(3) Interpretations from computational simulations. The authors performed computational simulations on the VP3 structure to infer how the protein might interact with membranes. Such computational approaches are powerful hypothesis-generating tools. However, additional biochemical evidence beyond what is presented would be required to support the authors' claims that they "unveiled a two-stage modular mechanism" for VP3-PI3P interactions (see lines 55-59). Moreover, given the biochemical data presented for R_200_D VP3, it was surprising that the authors did not perform computational simulations on this mutant. The inclusion of such an experiment would help tie together the in vitro and in silico data and strengthen the manuscript.

We acknowledge that the wording used in the previous version of the manuscript may have overstated the "unveiling" of the two-stage binding mechanism of VP3. Our intention was to propose a potential mechanism, that is consistent both with the biophysical experiments and the molecular simulations. In the revised version of the manuscript, we have tempered these claims and framed them more appropriately.

Regarding the simulations for the R_200_D VP3 mutant, these simulations were indeed performed and included in the original manuscript as part of Figure S14 in the Supplementary Information. However, we realize that this was not sufficiently emphasized in the main text, an oversight on our part. We have now revised the manuscript to highlight these results more clearly.

Additionally, to further strengthen the connection between experimental and simulation trends, we have now included a new figure in the Supplementary Information (Figure S15). This figure depicts the binding energy of VP3 ΔNt and two of its mutants, VP3 ΔNt R_200_D and VP3 ΔNt P2 Mut, as a function of salt concentration. The results show that as the number of positively charged residues in VP3 is systematically reduced, the binding of the protein to the membrane becomes weaker. The effect is more pronounced at lower salt concentrations, which highlights the weight of electrostatic forces on the adsorption of VP3 on negatively charged membranes. Please, see Supplementary Information (Figure S15).

**Reviewer #3 (Public Review):**
Summary:Infectious bursal disease virus (IBDV) is a birnavirus and an important avian pathogen. Interestingly, IBDV appears to be a unique dsRNA virus that uses early endosomes for RNA replication that is more common for +ssRNA viruses such as for example SARS-CoV-2.This work builds on previous studies showing that IBDV VP3 interacts with PIP3 during virus replication. The authors provide further biophysical evidence for the interaction and map the interacting domain on VP3.Strengths:Detailed characterization of the interaction between VP3 and PIP3 identified R_200_D mutation as critical for the interaction. Cryo-EM data show that VP3 leads to membrane deformation.Weaknesses:The work does not directly show that the identified R_200_ residues are directly involved in VP3-early endosome recruitment during infection. The majority of work is done with transfected VP3 protein (or in vitro) and not in virus-infected cells. Additional controls such as the use of PIP3 antagonizing drugs in infected cells together with a colocalization study of VP3 with early endosomes would strengthen the study.In addition, it would be advisable to include a control for cryo-EM using liposomes that do not contain PIP3 but are incubated with HIS-VP3-FL. This would allow ruling out any unspecific binding that might not be detected on WB.The authors also do not propose how their findings could be translated into drug development that could be applied to protect poultry during an outbreak. The title of the manuscript is broad and would improve with rewording so that it captures what the authors achieved.

In previous works from our group, we demonstrated the crucial role of the VP3 P2 region in targeting the early endosomal membranes and for viral replication, including the use of PI3K inhibitors to deplete PI3P, showing that both the control RFP-2xFYVE and VP3 lost their ability to associate with the early endosomal membranes and reduces the production of an infective viral progeny (*J Virol*. 2018 May 14;92(11):e01964-17, doi: 10.1128/jvi.01964-17; *J Virol*. 2021 Feb 24;95(6):e02313-20, doi: 10.1128/jvi.02313-20). In the present work, to further characterize the role of R_200_ in binding to early endosomes and for viral replication, we show that: (i) the transfected VP3 R_200_D protein loses the ability to bind to early endosomes in immunofluorescence assays (Figure 2E and Figure 3); (ii) the recombinant His-VP3 FL R_200_D protein loses the ability to bind to liposomes PI3P(+) in co-flotation assays (Figure 4A); and, (iii) the mutant virus R_200_D loses replication capacity (Figure 4C).

Regarding the cryo-electron microscopy observation, we verified that there is no binding of gold particles to liposomes PI3P(-) when they are incubated solely with the gold-particle reagent, or when they are pre-incubated with the gold-particle reagent with either His-2xFYVE or His-VP3 FL. We have incorporated a new panel in Figure 1C showing a representative image of these results. Please, see lines 143-144 in the revised version of our manuscript and our revised version of Figure 1C.

We have replaced the title of the manuscript by a more specific one. Thus, our current is " On the Role of VP3-PI3P Interaction in Birnavirus Endosomal Membrane Targeting".

Regarding the question of how our findings could be translated into drug development, indeed, VP3-PI3P binding constitutes a good potential target for drugs that counteract infectious bursal disease. However, we did not mention this idea in the manuscript, first because it is somewhat speculative and second because infected farms do not implement any specific treatment. The control is based on vaccination.

**Recommendations for the authors:**

**Reviewer #1 (Recommendations For The Authors):**
Critical issues to address:(1) The citations in the important paragraph on lines 101-5 are not identifiable. These references are described as showing that VP3 is associated with EEs via P2 and PI3P, which is basically what this paper also shows. The significant advance here is unclear.

We apologize for this mistake. These citations are identifiable in the revised version of the manuscript (lines 100-105). As mentioned before, in this manuscript we present biochemical and biophysical details that have not been reported before about how VP3 connects with early endosomes, showing that it interacts directly with the PI3P. Additionally, we have now identified a critical residue in VP3 P2—the R_200_—for binding to PI3P and its key role in the viral life cycle. Furthermore, the molecular dynamics simulations helped us come up with a mechanism for VP3 to connect with PI3P in early endosomes. This constitutes a big step forward in our understanding of how these "*non-canonical*" viruses replicate.

(2) Even if all the claims were to be clearly supported through major revamping, authors should make the significance of knowing that this protein binds to early endosomes through PI3P more clear?

Thank you for the recommendation, which aligns with a similar suggestion from Reviewer #2. In response, we have revised the significance paragraph to emphasize the mechanistic aspects of our findings. Please refer to lines 62–67 in the revised manuscript.

(3) Flotation assay shows binding, but this is not quantitative. An estimate of a Kd would be useful. BLI experiments suggest that half of the binding disappears at 0.5 mM, implying a very low binding affinity.

We agree with the reviewer that our biophysical and molecular simulation results suggest a specific but weak interaction of VP3 with PI3P bearing membranes. Indeed, our previous version of the manuscript already contained a paragraph in this regard. Please, see lines 323-332 in the revised version of the manuscript.

From a biological point of view, a low binding affinity of VP3 for the endosomes may constitute an advantage for the virus, in the sense that its traffic through the endosomes may be short lived during its infectious cycle. Indeed, VP3 has been demonstrated to be a "multifunctional" protein involved in several processes of the viral cycle (detailed in lines 84-90), and in our laboratory we have shown that the Golgi complex and the endoplasmic reticulum are organelles where further viral maturation occurs. Taking all of this into account, a high binding affinity of VP3 for endosomes could result in the protein becoming trapped on the endosomal membrane, potentially hindering the progression of the viral infection within the host cell.

(4) There are some major internal inconsistencies in the data: Figure 1B quantifies VP3-FL T/B ratio ~4 (which appears inconsistent with the image shown, as the T lanes are much lighter than the B) whereas apparently the same experiment in Figure 1G shows it to be ~0.6. With the error bars shown, these results would appear dramatically different from each other, despite supposedly measuring the same thing. The same issue with the FYVE domain between Figures 1A and 4A.

We appreciate the reviewer’s comment, as it made us aware of an error in Figure 1B. There, the mean value for the VP3-FL Ts/B ratio is 3.0786 for liposomes PI3P(+) and 0.4553 for liposomes PI3P(-) (Please, see the new bar graph on Figure 1B). This may have occurred because, due to the significance of these experiments, we performed multiple rounds of quantification in search of the most suitable procedure for our observations, leading to a mix-up of data sets. Anyway, it’s possible that these corrected values still seem inconsistent given that T lanes are much lighter than the B for VP3-FL in the image shown. Flotation assays are quite labor-intensive and, at least in our experience, yield fairly variable results in terms of quantification. To illustrate this point, the following image shows the three experiments conducted for Figure 1B, where it is clear that, despite producing visually distinct images, all three yielded the same qualitative observation. For Figure 1B, we chose to present the results from experiment #2. However, all three experiments contributed to a Ts/B ratio of 3.0786 for His-VP3 FL, which may account for the apparent inconsistency when focusing solely on the image in Figure 1B.

We acknowledge that, at first glance, some inconsistencies may appear in the results, and we have thoroughly discussed the best approach for quantification. However, we believe the observations are robust in terms of reproducibility and reliable, as the VP3-PI3P interaction was consistently validated by comparison with liposomes lacking PI3P, where no binding was observed.

(5) Comparison of PA (or PI) to PI3P at the same molar concentration is inappropriate because PI3P has at least double charge. The more interesting question about specificity would be whether PI45P2 (or even better PI35P2) binds or not. Without this comparison, no claim to specificity can be made.

For us, "specificity" refers to the requirement of a phosphoinositide in the endosomal membrane for VP3 binding. Phosphoinositides have a conspicuous distribution among cellular compartments, and knowing that VP3 associates with early endosomes, our specificity assays aimed to demonstrate that PI3P is strictly required for the binding of VP3. To validate this, we used PI (lacking the phosphate group) and PA (lacking the inositol group) despite their similar charges. In spite of the potential chemical interactions between VP3 and various phosphoinositides, our experimental results suggest that the virus specifically targets endosomal membranes by binding to PI3P, a phosphoinositide present only in early endosomes.

That said, we agree with the reviewer’s point and consider adequate to smooth our specificity claim in the manuscript as follows: “We observed that His-VP3 FL bound to liposomes PI3P(+), but not to liposomes PA or PI, reinforcing the notion that a phosphoinositide is required since neither a single negative charge nor an inositol ring are sufficient to promote VP3 binding to liposomes (SI Appendix, Fig. S2)” (Lines 136-139).

(6) In the EM images, many of the gold beads are inside the vesicles. How do they cross the membranes?

They do not cross the membrane. Our EM images are two-dimensional projections, meaning that the gold particles located on top or beneath the plane appear to be inside the liposome.

(7) Images in Figure 2D are very low quality and do not show the claimed difference between any of the mutants. All red signal looks basically cytosolic in all images. It is not clear what criteria were used for the quantification in Figure 2E. The same issue is in Figure 2E, where no red WT puncta are observable at all. Consistently, there is minimal colocalization in the quantification in Figure S3, which appears to show no significant differences between any of the mutants, in direct contradiction to the claim in the manuscript.

We apologize for the poor quality of panels in Figures 2D and 2E. Unfortunately, this was due to the PDF conversion of the original files. Please, check the high-quality version of Figure 2. As suggested by reviewers #2 and #3, we have incorporated zoomed panels, which help the reader to better see the differences in distribution.

As mentioned in the legend to Figure 2, the quantification in Figure 2D was performed by calculating the percentage of cells with punctuated fluorescent red signal (showing VP3 distribution) for each protein. The data were then normalized to the P2 WT protein, which is the VP3 wild type.

Figure S3 certainly shows a tendency which positively correlates with the results shown in Figure 3, where we used FYVE to detect PI3P on endosomes and observed significantly less co-localization when VP3 bears its P2 region all reversed or lacks the R_200_

(8) The only significant differences in colocalization are in Figure 3B, whose images look rather dramatically different from the rest of the manuscript, leading to some concern about repeatability. Also, it is unclear how colocalization is quantified, but this number typically cannot be above 1. Finally, it is unclear what is being colocalized here: with three fluorescent components, there are 3 possible binary colocalizations and an additional ternary colocalization.

We thank the reviewer for pointing out those aspects related to Figure 3. The experiments performed for Figure 3B were conducted by a collaborator abroad handling the purified GST-2xFYVE, which recognizes endogenous PI3P, while the rest of the cell biology experiments were conducted in our laboratory in Argentina. This is why they are aesthetically different. We have made an effort in homogenizing the way they look for the revised version of the manuscript. Please, see the new version of Figure 3.

For quantification of the co-localization of VP3 and EGFP-2xFYVE (Figure 3A), the Manders M2 coefficient was calculated out of approximately 30 cells per construct and experiment. The M2 coefficient, which reflects co-localization of signals, is defined as the ratio of the total intensities of magenta image pixels for which the intensity in the blue channel is above zero to the total intensity in the magenta channel. JACoP plugin was utilized to determine M2. For VP3 puncta co-distributing with EEA1 and GST-FYVE (Figure 3B), the number of puncta co-distributing for the three signals was manually determined out of approximately 40 cells per construct and experiment per 200 µm². We understand that Manders or Pearson coefficients, typically ranging between 0 and 1, is the most commonly used method to quantify co-localizing immunofluorescent signals; however, this “manual” method has been used and validated in previous published manuscripts [Figures 3 and 7 from (Morel et al., 2013); Figure 7 in (Khaldoun et al., 2014); and Figure 4 in (Boukhalfa et al., 2021)].

(9) SegA/B plasmids are not introduced, and it is not clear what these are or how this assay is meant to work. Where are the foci forming units in the images of Figure 4C? How does this inform on replication? Again, this assay is not quantitative, which is essential here: does the R_200_ mutant completely kill activity (whatever that is here)? Or reduce it somewhat?

We apologize for the missing information. Segments A and B are basically the components of the IBDV reverse genetics system. For their construction, we used a modification of the system described by Qi and coworkers (Qi et al., 2007), in which the full length sequences of the IBDV RNA segments A and B, flanked by a hammerhead ribozyme at the 5’-end and the hepatitis delta ribozyme at the 3’-end, were expressed under the control of an RNA polymerase II promoter within the plasmids pCAGEN.Hmz.SegA.Hdz (SegA) and pCAGEN.Hmz.SegB.Hdz (SegB). For this specific experiment we generated a third plasmid, pCAGEN.Hmz.SegA.R_200_D.Hdz (SegA.R_200_D), harboring a mutant version of segment A cDNA containing the R_200_D substitution. Then, QM7 cells were transfected with the plasmids SegA, SegB or Seg.R_200_D alone (as controls) or with a mixture of plasmids SegA+SegB (wild type situation) or SegA.R_200_D+SegB (mutant situation). At 8 h post transfection (p.t.), when the new viruses have been able to assemble starting from the two segments of RNA, the cells were recovered and re-plated onto fresh non-transfected cells for revealing the presence (or not) of infective viruses. At 72 h post-plating, the generation of foci forming units (FFUs) was revealed by Coomassie staining. As expected, single-transfections of SegA, SegB or Seg.R_200_D did not produce FFUs and, as shown in Figure 4C, the transfection of SegA+SegB produced detectable FFUs (the three circles in the upper panel) while no FFUs (the three circles in the lower panel) were detected after the transfection of SegA.R_200_D+SegB (Figure 4C). This system is quantitative, since the FFUs detected 72 h post-plating are quantifiable by simply counting the FFUs. However, since no FFUs were detected after the transfection of SegA.R_200_D+SegB, evidenced by a complete monolayer of cells stained blue, we did not find any sense in quantifying. In turn, this drastic observation indicates that viruses bearing the VP3 R_200_D mutation lose their replication ability (is “dead”), demonstrating its crucial role in the infectious cycle.

We agree with the reviewer that a better explanation was needed in the manuscript, so we have incorporated a paragraph in the results section of our revised version of the manuscript (lines 209-219).

(10) Why pH 8 for simulation?

The Molecular Theory calculations were performed at pH 8 for consistency with the experimental conditions used in our biophysical assays. These biophysical experiments were also performed at pH 8, following the conditions established in the original study where VP3 was first purified for crystallization (DOI: 10.1016/j.str.2007.10.023).

(11) There is minimal evidence for the sequential binding model described in the abstract. The simulations do not resolve this model, nor is truly specific PI3P binding shown.

In response to your concerns, we would like to emphasize that our simulations provide robust evidence supporting the two more important aspects of the sequential binding model: (1) Membrane Approach: In all simulations, VP3 consistently approaches the membrane via its positively charged C-terminal (Ct) region. (2) PI3P Recruitment: Once the protein is positioned flat on the membrane surface, PI3P is unequivocally recruited to the positively charged P2 region. The enrichment of PI3P in the proximity to the protein is clearly observed and has been quantified via radial distribution functions, as detailed in the manuscript and supplementary material.

While we understand that opinions may vary on the sufficiency of the data to fully validate the model, we believe the results offer meaningful insights into the proposed binding mechanism. That said, we acknowledge that the specificity of VP3 binding may not be restricted solely to PI3P but could extend to phosphoinositides in general. To address this, we performed the new set of co-flotation experiments which are discussed in detail in our response to point 5.

**Reviewer #2 (Recommendations For The Authors):**
(1) Line 1: Consider changing the title to better reflect the mostly biochemical and computational data presented in the paper: "Mechanism of Birnavirus VP3 Interactions with PI3P-Containing Membranes". There are no data to show hijacking by a virus presented.

We appreciate this recommendation, which was also expressed by reviewer #3. Additionally, we thank for the suggested title. We have replaced the title of the manuscript by a more specific one. Thus, our current is

"On the Role of VP3-PI3P Interaction in Birnavirus Endosomal Membrane Targeting".

(2) Lines 53-54 and throughout: Consider rephrasing "demonstrate" to "validate" to give credit to Gimenez et al., 2018, 2022 for discovery.

Thanks for the suggestion. We have followed it accordingly. Please see line 52 from our revised version of the manuscript.

(3) Line 56-59 and throughout: Consider tempering and rephrasing these conclusions that are based mostly on computational data. For example, change "unveil" to "suggest" or another term.

We have now modified the wording throughout the manuscript.

(4) The abstract could also emphasize that this study sought to map the resides within VP3 that are important for P13P interaction.

Thanks for the suggestion. We have followed it accordingly. Please, see lines 53-55 from our revised version of the manuscript.

(5) Lines 63-69: This Significance paragraph seems tangential. The findings in this paper aren't at all related to the evolutionary link between birnaviruses and positive-strand RNA viruses. The significance of the work for me lies in the deep biochemical/biophysical insights into how a viral protein interacts with membranes to nucleate its replication factory.

We have re-written the significance paragraph highlighting the mechanistic aspect of our findings. Please, see lines 62-67 in our revised version of the manuscript.

(6) Line 74: Please define "IDBV" abbreviation.

We apologize for the missing information. We have defined the IBDV abbreviation in our revised version of the manuscript (please, see line 73).

(7) Line 88: Please define "pVP2" abbreviation.

We apologize for the missing information. We have defined the pVP2 abbreviation in our revised version of the manuscript (please, see line 87).

(8) Lines 101-105: Please change references (8, 9, 10) to be consistent with the rest of the manuscript (names, year).

We apologize for this mistake. These citations are identifiable and consistent in the revised version of the manuscript (lines 100-105).

(9) Line 125: For a broad audience, consider explaining that recombinant His-2xFYVE domain is known to exhibit PI3P-binding specificity and was used as a positive control.

Thanks for the recommendation. We have incorporated a brief explanation supporting the use of His-2xFYVE as a positive control in our revised version of the manuscript. Please, see lines 127-129.

(10) Lines 167-171: The quantitative data in Figure S3 shows that there was a non-significant co-localization coefficient of the R_200_D mutant. For transparency, this should be stated in the Results section when referenced.

We agree with this recommendation. We have clearly mentioned it in the revised version of the manuscript. Please, see lines 177-179. Also, we have referred this fact when introducing the assays performed using the purified GST-2xFYVE, shown in Figure 3. Please, see lines 182-184.

(11) Lines 156 and 173: These Results section titles have nearly identical wording. Consider rephrasing to make it distinct.

We agree with the reviewer’s observation. In fact, we sought to do it on purpose as for them to be a “wordplay”, but we understand that could result in a awkwarded redundancy. So, in the revised version of the manuscript, both titles are:

Role of VP3 P2 in the association of VP3 with the EE membrane (line 163).

VP3 P2 mediates VP3-PI3P association to EE membranes (line 182).

(12) Line 194: Is it alternatively possible that the R_200_D mutant lost its capacity to dimerize, and that in turn impacted PI3P interaction?

Thanks for the relevant question. VP3 was crystallized and its structure reported in (Casañas et al., 2008) (DOI: 10.1016/j.str.2007.10.023). In that report, the authors showed that the two VP3 subunits associate in a symmetrical manner by using the crystallographic two-fold axes. Each subunit contributes with its 30% of the total surface to form the dimer, with 81 interprotomeric close contacts, including polar bonds and van der Waals contacts. The authors identified the group of residues involved in these interactions, among which the R_200_ is not included. Addittionally, the authors determined that the interface of the VP3 dimer in crystals is biologically meaningful (not due to the crystal packing).

To confirm that the lack of binding was not due to misfolding of the mutant, we compared the circular dichroism spectra of mutant and wild type proteins, without detecting significant differences (shown in Figure 4B). These observations do not exclude the possibility mentioned by the reviewer, but constitute solid evidences, we believe, to validate our observations.

(13) Lines 231-243: Consider changing verbs to past tense (i.e., change "is" to "was") for the purposes of consistency and tempering.

Thanks for the recommendation, we have proceeded as suggested. Please, see lines 249-262 in our revised version of the manuscript.

(14) Lines 306-308: Is there any information about whether it is free VP3 (v. VP3 complexed in RNP) that binds to membrane? I am just trying to wrap my head around how these factories form during infection.

Thanks for pointing this out. We first observed that in infected cell, all the components of the RNPs [VP3, VP1 (the viral polymerase) and the dsRNA] were associated to the endosomes. Since by this moment it had been already elucidated that VP3 "wrapped" de dsRNA within the RNPs (Luque et al., 2009) (DOI: 10.1016/j.jmb.2008.11.029), we sought that VP3 was most probably leading this association. We answered yes after studying its distribution, also endosome-associated, when ectopically expressed. These results were published in (Delgui et al., 2013) (DOI: 10.1128/jvi.03152-12).

Thus, in our subsequent studies, we have worked with both, the infection-derived or the ectopically expressed VP3, to advance in elucidating the mechanism by which VP3 hijacks the endosomal membranes and its relevancy for viral replication, reported in this current manuscript.

(15) Lines 320-334: This last paragraph discussing evolutionary links between birnaviruses and positive-strand RNA viruses seems tangential and distracting. Consider reducing or removing.

Thanks for highlighting this aspect of our work. Maybe difficult to follow, but in the context of other evidences reported for the *Birnaviridae* family of viruses, we strongly believe that there is an evolutionary aspect in having observed that these dsRNA viruses replicate associated to membranous organelles, a hallmark of +RNA viruses. However, we agree with the reviewer that this might not be the main point of our manuscript, so we reduced this paragraph accordingly. Please, see lines 358-367 in our revised version of the manuscript.

(16) Lines 322-324: Change "RdRd" to "RdRp" if keeping paragraph.

Thanks. We have corrected this mistake in lines 360 and 361.

(17) Figures 1A, 1B, and throughout: Again, please check and explain protein sizes and amounts. This would improve the clarity of the manuscript.

All our flotation assays were performed using 1 mM concentration of purified protein in a final volume of 100 mL (mentioned in M&M section). The complete fusion protein His-2xFYVE (shown in Figs. 1A and 4A left panel) is 954 base pairs-long and contains 317 residues (~35 kDa). The complete fusion protein His-VP3 FL (shown in Figs. 1B and 1G left panel) is 861 base pairs-long and contains 286 residues (~32 kDa). The complete fusion protein His-VP3 DCt (shown in Fig. 1G, right panel) is 753 bp-long and contains 250 residues (~28 kDa). The complete fusion protein His-VP3 FL R_200_D (shown in Fig. 4A right panel) is 861 bp-long and contains 286 residues (~32 kDa). This latter information was incorporated in our revised version of the manuscript. Please, see lines 381-382, 396-397 and 399-400 from the M&M section, and lines in the corresponding figure legends.

(18) Figures 1B and 1G show different results for PI3P(+) membranes. I see protein associated with the top fraction in 1B, but I don't see any such result in 1G.

As already mentioned, liposome-based methods, such as the co-flotation assay, are well-established and widely regarded as the preferred approach for studying protein-phosphoinositide interactions. However, this approach is rather qualitative, as density gradient separation reveals whether the protein is located in the top fractions (bound to liposomes) or the bottom fractions (unbound). Our quantifications aim to demonstrate differences in the bound fraction between liposome populations with and without PI3P. Given the setting of the co-flotation assays, each protein-liposome system [2xFYVE-PI3P(-), 2xFYVE-PI3P(+), VP3-PI3P(-), or VP3-PI3P(+)] is assessed separately, and even if the conditions are homogeneous, it’s not surprising to observe differences in the protein level between each one. Indeed, the revised version of the manuscript include a membrane for Figure 1G, were His-VP3 FL associated with the top fraction is more clear. Please, see the new version of Figure 1G.

(19) Figure 1C: Please include cryo-EM images of the liposome PI3P(-) variables to assess the visual differences of the liposomal membranes under these conditions.

Thanks for the recommendation. it has been verified that there is no binding of gold particles to liposomes PI3P(-) when they are incubated solely with the gold-particle reagent, or when they are pre-incubated with the gold-particle reagent with either His-2xFYVE or His-VP3 FL. We have incorporated a new panel in Figure 1C showing a representative image of these results. Please, see lines 143-144 in the revised version of our manuscript and our revised version of Figure 1C.

(20) Figures 2D, 2E, and 3A: The puncta are not obvious in these images. Consider adding Zoomed panels.

We apologize for this aspect of Figures 2 and 3, also highlighted by reviewer #1. We believe that this was due to the low quality resulting from the PDF conversion of the original files. For Figure 3A, we have homogenized its aspect with those from 3B. Regarding Figure 2, we have incorporated zoomed panels, as suggested. Please, see the revised versions of both Figures.

(21) Figure 4A: There is almost no protein in the control PI3P(+) blot. Why? Also, the quantification shows no significant membrane association for this control. This result is different from Figure 1A and very confusing (and concerning).

We apologize for the confusion. We replaced membranes for Figure 4A (left panel) with more similar band intensities to that shown in Figure 1A. Please, visit our new version of Figure 4. The quantification shows no significant difference in the association to liposomes PI3P(+) compared to liposomes PI3P(+); it’s true and this is due to, once more, the intrinsically lack of homogeneity of co-flotation assays. However, this one shown in Figure 4A is a redundant control (has been shown in Figure 1A) and we believe that the new membrane is qualitative eloquent.

**Reviewer #3 (Recommendations For The Authors):**
(1) Overall, the title is general and does not summarize the study. I recommend making the title more specific. The current title is better suited for a review as opposed to a research article. This study provides further biophysical details on the interaction. This should be reflected in the title.

We appreciate this recommendation, which was also expressed by reviewer #2. We have chosen a new title for the manuscript: “On the Role of VP3-PI3P Interaction in Birnavirus Endosomal Membrane Targeting”.

(2) References 8,9,10 are important but they were not correctly cited in the work, this should be corrected.

We apologize for this mistake. These citations are identifiable in our revised version of the manuscript. See lines 100-105.

(3) Flotation experiments and cryo-EM convincingly show that VP3 binds to membranes in a PIP3-dependent manner. However, it would be advisable to include a control for cryo-EM using liposomes that do not contain PIP3 but are incubated with HIS-VP3-FL. This would allow us to rule out any unspecific binding that might not be detected on WB.

Thanks for the advice, also given by reviewer #2. We confirmed that no gold particles were bound on liposomes PI3P(-) even when incubated with the Ni-NTA reagent alone or pre-incubated with His-2xFYVE of His-VP3 FL. We have incorporated a new panel to Figure 1C showing a representative image of these results. Please, see lines 143-144 in the revised version of the manuscript and see the revised version of Figure 1C.

(4) It is not clear what is the difference between WB in B and WB in G. Figure 1G seems to show the same experiment as shown in B, is this a repetition? In both cases, plots next to WBs show quantification with bars, do they represent STD or SEM? Legend A mentions significance p>0.01 (**) but the plot shows ***. This should be corrected.

The Western blot membrane in Figure 1B shows the result of co-flotation assay using His-VP3 FL protein, while the Western blot membrane in Figure 1G (left panel) shows a co-flotation assay using His-VP3 FL protein as a positive control. In another words, in 1B the His-VP3 FL protein is the question while in 1G (left panel) it’s the co-flotation positive control for His-VP3 DCt. The bar plots next to Western blots show quantification, the mean and the STD. Thanks for highlighting this inconsistency. We have now corrected it on the revised version of the manuscript.

(5) It would be useful to indicate positively charged residues and P2 on the AF2 predicted structure in Fig 1.

These are indicated in panels A and B of Figure 2.

(6) Figure 1 legend: Change cryo-fixated liposomes to cryo-fixation or better to "liposomes were vitrified". There is a missing "o" in the cry-fixation in the methods section.

Thanks for the recommendation. We have modified Figure 1. legend to "liposomes were vitrified" (line 758), and fixed the word cryo-fixation in the methods section (line 512).

(7) Figure 2B. It is not clear how the punctated phenotype was unbiasedly characterized (Figure 2D). I see no difference in the representative images. Magnified images should be shown. This should be measured as colocalization (Pearson's and Mander's coefficient) with an early endosomal marker Rab5. Perhaps this figure could be consolidated with Figure 3.

Unfortunately, the lack of clarity in Figure 2D was due to the PDF conversion of the original files. Please, observe the high-quality original image above in response to reviewer #1, where we have additionally included zoomed panels, as also suggested by the other reviewers. For quantification of the co-localization of VP3 and either EGFP-Rab5 orEGFP-2xFYVE, the Manders M2 coefficient was calculated out of approximately 30 cells per construct and experiment and were shown in Figure S3 and Figure 3A, respectively, in our previous version of the manuscript.

(8) PIP3 antagonist drugs should be used to further substantiate the results. If PIP3 specifically recruits VP3, this interaction should be abolished in the presence of PIP3 drug and VP3 should show a diffused signal.

We certainly agree with this point. These experiments were performed and the results were reported in (Gimenez et al., 2020). Briefly, in that work, we blocked the synthesis of PI3P in QM7 cells in a stable cell line overexpressing VP3, QM7-VP3, with either the pan-PI3Kinase (PI3K) inhibitor LY294002, or the specific class III PI3K Vps34 inhibitor Vps34-IN1. In Figure 4, we showed that 98% of the cells treated with these inhibitors had the biosensor GFP-2FYVE dissociated from EEs, evidencing the depletion of PI3P in EEs (Figure 4A). In QM7-VP3 cells, we showed that the depletion of PI3P by either inhibitor caused the dissociation of VP3 from EEs and the disaggregation of VP3 puncta toward a cytosolic distribution (Figure 4B). Moreover, since this observation was crucial for our hipothesis, these results were further confirmed with an alternative strategy to deplete PI3P in EEs. We employed a system to inducibly hydrolyze endosomal PI3P through rapamycin-induced recruitment of the PI3P-myotubularin 1 (MTM1) to endosomes in cells expressing MTM1 fused to the FK506 binding protein (FKBP) and the rapamycin-binding domain fused to Rab5, using the fluorescent proteins mCherry-FKBP-MTM1 and iRFP-FRB-Rab5, as described in (Hammond et al., 2014). These results, shown in Figures 5, 6 and 7 in the same manuscript, further reinforced the notion that PI3P mediates and is necessary for the association of VP3 protein with EEs.

(9) The authors should show the localization of VP3 in IBDV-infected cells and treat cells with PI3P antagonists. The fact that R_200_ is not rescued does not necessarily mean that this is because of the failed interaction with PI3P. As the authors wrote in the discussion: VP3 bears multiple essential roles during the viral life cycle (line 305).

Indeed, after having confirmed that the VP3 lost its localization associated to the endosomes after the treatment of the cells with PI3P antagonists, we demonstrated that depletion of PI3P significantly reduced the production of IBDV progeny. For this aim, we used two approaches, the inhibitor Vps34-IN1 and an siRNA against VPs34. In both cases, we observed a significantly reduced production of IBDV progeny (Figures 9 and 10). Specifically related to the reviewer’s question, the localization of VP3 in IBDV-infected cells and treated with PI3P antagonists was shown and quantified in Figure 9a.

(10) Could you provide adsorption-free energy profiles and MD simulations also for the R_200_ mutant?

Following the reviewer’s suggestion, we have added a new figure to the supplementary information (Figure S15). Instead of presenting a full free-energy profile for each protein, we focused on the adsorption free energy (i.e., the minimum of the adsorption free-energy profile) for VP3 ΔNt and its mutants, VP3 ΔNt R_200_D and VP3 ΔNt P2 Mut, as a function of salt concentration. The aim was to compare the adsorption free energy of the three proteins and evaluate the effect of electrostatic forces on it, which become increasingly screened at higher salt concentrations. As shown in the referenced figure, reducing the number of positively charged residues from VP3 ΔNt to VP3 ΔNt P2 Mut systematically weakens the protein’s binding to the membrane. This effect is particularly pronounced at lower salt concentrations, underscoring the importance of electrostatic interactions in the adsorption of the negatively charged VP3 onto the anionic membrane.

(11) Liposome deformations in the presence of VP3 are interesting (Figure 6G), were these also observed in Figure 1C?

Good question. The liposome deformations in the presence of VP3 shown in Figure 6G were a robust observation since, as mentioned, it was detectable in 36% of the liposomes PI3P(+), while they were completely absent in PI3P(-) liposomes. However, and unfortunately, the same deformations were not detectable in experiments performed using gold particles shown in Figure 1C. In this regard, we think that it might be possible that the procedure of gold particles incubation itself, or even the presence of the gold particles in the images, would somehow “mask” the deformations effect.

Bibliography

Boukhalfa A, Roccio F, Dupont N, Codogno P, Morel E. 2021. The autophagy protein ATG16L1 cooperates with IFT20 and INPP5E to regulate the turnover of phosphoinositides at the primary cilium. *Cell Rep* 35:109045. doi:10.1016/j.celrep.2021.109045

Casañas A, Navarro A, Ferrer-Orta C, González D, Rodríguez JF, Verdaguer N. 2008. Structural Insights into the Multifunctional Protein VP3 of Birnaviruses. *Structure* 16:29–37. doi:10.1016/j.str.2007.10.023

Delgui LR, Rodriguez JF, Colombo MI. 2013. The Endosomal Pathway and the Golgi Complex Are Involved in the Infectious Bursal Disease Virus Life Cycle. *J Virol* 87:8993–9007. doi:10.1128/JVI.03152-12

Gimenez MC, Issa M, Sheth J, Colombo MI, Terebiznik MR, Delgui LR. 2020. Phosphatidylinositol 3-Phosphate Mediates the Establishment of Infectious Bursal Disease Virus Replication Complexes in Association with Early Endosomes. *J Virol* 95:e02313-20. doi:10.1128/jvi.02313-20

Hammond GRV, Machner MP, Balla T. 2014. A novel probe for phosphatidylinositol 4-phosphate reveals multiple pools beyond the Golgi. *J Cell Biol* 205:113–126. doi:10.1083/jcb.201312072

Khaldoun SA, Emond-Boisjoly MA, Chateau D, Carrière V, Lacasa M, Rousset M, Demignot S, Morel E. 2014. Autophagosomes contribute to intracellular lipid distribution in enterocytes. *Mol Biol Cell* 25:118. doi:10.1091/mbc.E13-06-0324

Luque D, Saugar I, Rejas MT, Carrascosa JL, Rodríguez JF, Castón JR. 2009. Infectious Bursal Disease Virus: Ribonucleoprotein Complexes of a Double-Stranded RNA Virus. *J Mol Biol* 386:891–901. doi:10.1016/j.jmb.2008.11.029

Morel E, Chamoun Z, Lasiecka ZM, Chan RB, Williamson RL, Vetanovetz C, Dall’Armi C, Simoes S, Point Du Jour KS, McCabe BD, Small SA, Di Paolo G. 2013. Phosphatidylinositol-3-phosphate regulates sorting and processing of amyloid precursor protein through the endosomal system. *Nature Communications 2013 4:1* 4:1–13. doi:10.1038/ncomms3250

Qi X, Gao Y, Gao H, Deng X, Bu Z, Wang Xiaoyan, Fu C, Wang Xiaomei. 2007. An improved method for infectious bursal disease virus rescue using RNA polymerase II system. *J Virol Methods* 142:81–88. doi:10.1016/j.jviromet.2007.01.021